# DyG2Vec: Representation Learning for Dynamic Graphs with Self-Supervision

## Abstract

The challenge in learning from dynamic graphs for predictive tasks lies in extracting fine-grained temporal motifs from an ever-evolving graph. Moreover, task labels are often scarce, costly to obtain, and highly imbalanced for large dynamic graphs. Recent advances in self-supervised learning on graphs demonstrate great potential, but focus on static graphs. State-of-the-art (SoTA) models for dynamic graphs are not only incompatible with the self-supervised learning (SSL) paradigm but also fail to forecast interactions beyond the very near future. To address these limitations, we present DyG2Vec, an SSL-compatible, efficient model for representation learning on dynamic graphs. DyG2Vec uses a window-based mechanism to generate task-agnostic node embeddings that can be used to forecast future interactions. DyG2Vec significantly outperforms SoTA baselines on benchmark datasets for downstream tasks while only requiring a fraction of the training/inference time. We adapt two SSL evaluation mechanisms to make them applicable to dynamic graphs and thus show that SSL pre-training helps learn more robust temporal node representations, especially for scenarios with few labels.

## 1 Introduction

Graph Neural Networks (GNNs) have recently found great success in representation learning for complex networks of interactions, as present in recommendation systems, transaction networks, and social media (Wu et al., 2020; Zhang et al., 2019; Qiu et al., 2018). However, most approaches ignore the dynamic nature of graphs encountered in many real-world domains. Dynamic graphs model complex, time-evolving interactions between entities (Kazemi et al., 2020; Skarding et al., 2021; Xue et al., 2022). Multiple works have revealed that real-world dynamic graphs possess fine-grained temporal patterns known as temporal motifs (Toivonen et al., 2007; Paranjape et al., 2017). For example, a simple pattern in social networks specifies that two users who share many friends are likely to interact in the future. A robust representation learning approach must be able to extract such temporal patterns from an ever-evolving dynamic graph in order to make accurate predictions.

Self-Supervised Representation Learning (SSL) has shown promise in achieving competitive performance for different data modalities on multiple predictive tasks (Liu et al., 2021). Given a large corpus of unlabelled data, SSL postulates that unsupervised pre-training is sufficient to learn robust representations that are predictive for downstream tasks with minimal fine-tuning. However, it is important to specify a pre-training objective function that induces good performance for the downstream tasks. Contrastive SSL methods, despite their early success, rely heavily on negative samples, extensive data augmentation, and large batch sizes (Jing et al., 2022; Garrido et al., 2022). Non-contrastive methods address these shortcomings, incorporating information theoretic principles through architectural innovations or regularization methods. These closely resemble strategies employed in manifold learning and spectral embedding methods (Balestriero & LeCun, 2022). The success of such SSL methods on sequential data (Tong et al., 2022; Eldele et al., 2021; Patrick et al., 2021) suggests that one can learn rich temporal node embeddings from dynamic graphs without direct supervision.

SSL methods are attractive for dynamic graphs because it is often costly to generate ground truth labels. Contrastive approaches are very sensitive to the quality of the negative samples, and these are challenging to identify in dynamic graphs due to the temporal evolution of interactions and the

lack of semantic labels at the contextual level. As a result, it is desirable to explore non-contrastive techniques, but state-of-the-art models for dynamic graphs suffer from shortcomings that make them hard to adapt to SSL paradigms. First, they heavily rely on chronological training or a full history of interactions to construct predictions (Kumar et al., 2019; Xu et al., 2020; Rossi et al., 2020; Wang et al., 2021b). Second, the encoding modules either use inefficient message-passing procedures (Xu et al., 2020), memory blocks (Kumar et al., 2019; Rossi et al., 2020), or expensive random walk-based algorithms (Wang et al., 2021b) that are designed for edge-level tasks only. As a result, while SSL pre-training has been applied successfully for static graphs (Thakoor et al., 2022; Hassani & Khasahmadi, 2020; You et al., 2022), there has been limited success in adapting SSL pre-training to dynamic graphs.

In this work, we propose DyG2Vec [1], a novel encoder-decoder model for continuous-time dynamic graphs that benefits from a window-based architecture that acts a regularizer to avoid over-fitting. DyG2Vec is an efficient attention-based graph neural network that performs message-passing across structure and time to output task-agnostic node embeddings. Experimental results for 7 benchmark datasets indicate that DyG2Vec outperforms SoTA baselines on future link prediction and dynamic node classification in terms of performance and speed, particularly in medium- and long-range forecasting. The novelty of our model lies in its compatibility with SoTA SSL approaches. That is, we propose a joint-embedding architecture for DyG2Vec that can benefit from non-contrastive SSL. We adapt two evaluation protocols (linear and semi-supervised probing) to the dynamic graph setting and demonstrate that the proposed SSL pre-training is effective in the low-label regime.

## 2 RELATED WORK

**Self-supervised representation learning:** Multiple works explore learning visual representations without labels (see (Liu et al., 2021) for a survey). The more recent contrastive methods generate random views of images through data augmentations, and then force representations of positive pairs to be similar while pushing apart representations of negative pairs (Chen et al., 2020a; He et al., 2019). With the goal of attaining hard negative samples, such methods typically use large batch sizes Chen et al. (2020a) or memory banks (He et al., 2019; Chen et al., 2020b). Non-contrastive methods such as BYOL (Grill et al., 2020) and VICReg (Bardes et al., 2022) eliminate the need for negative samples through various techniques that avoid representation collapse (Jing et al., 2022). Recently, several SSL methods have been adapted to pre-train GNNs (Xie et al., 2022). Deep Graph Infomax (DGI) (Velickovic et al., 2019) and InfoGCL (Xu et al., 2021)) rely on mutual information maximization or information bottle-necking between patch-level and graph-level summaries. MV-GRL (Hassani & Khasahmadi, 2020) incorporates multiple views, and BGRL (Thakoor et al., 2022) adapts BYOL to graphs to eliminate the need for negative samples, which are often memory-heavy in the graph setting. The experiments demonstrate the high degree of scalability of non-contrastive methods and their effectiveness in leveraging both labeled and unlabeled data.

**Representation learning for dynamic graphs:** Early works on representation learning for continuous-time dynamic graphs typically divide the graph into snapshots that are encoded by a static GNN and then processed by an RNN module (Sankar et al., 2020; Pareja et al., 2020; Kazemi et al., 2020). Such methods fail to learn fine-grained temporal patterns at smaller timescales within each snapshot. Therefore, several RNN-based methods were introduced that sequentially update node embeddings as new edges arrive. JODIE (Kumar et al., 2019) employs two RNN modules to update the source and destination embeddings respectively of a arriving edge. DyRep (Trivedi et al., 2019) adds a temporal attention layer to take into account multi-hop interactions when updating node embeddings. TGAT (Xu et al., 2020) includes an attention-based message passing (AMP) architecture to aggregate messages from a historical neighborhood. TGN (Rossi et al., 2020) alleviates the expensive neighborhood aggregation of TGAT by using an RNN memory module to encode the history of each node. CaW (Wang et al., 2021b) extracts temporal patterns through an expensive procedure that samples temporal random walks and encodes them with an LSTM. This procedure must be performed for every prediction. In contrast to prior works, our method operates on a fixed window of history to generate node embeddings. Additionally, we do not recompute embeddings for every prediction, which allows for efficient computation and memory usage.

---

[1] We are going to open-source the code upon acceptance.

## 3 PROBLEM FORMULATION

A Continuous-Time Dynamic Graph (CTDG) $\mathcal{G} = (\mathcal{V}, \mathcal{E}, \mathcal{X})$ is a sequence of $E = |\mathcal{E}|$ interactions, where $\mathcal{X} = (X^V, X^E)$ is the set of input features containing the *node features* $X^V \in \mathbb{R}^{N \times D^V}$ and the *edge features* $X^E \in \mathbb{R}^{E \times D^E}$. $\mathcal{E} = \{e_1, e_2, \dots, e_E\}$ is the set of interactions. There are $N = |\mathcal{V}|$ nodes, and $D^V$ and $D^E$ are the dimensions of the node and edge feature vectors, respectively. An edge $e_i = (u_i, v_i, t_i, m_i)$ is an interaction between any two nodes $u_i, v_i \in \mathcal{V}$, with $t_i \in \mathbb{R}$ being a continuous timestamp, and $m_i \in X^E$ an edge feature vector. For simplicity, we assume that the edges are undirected and ordered by time (i.e., $t_i \leq t_{i+1}$). A temporal sub-graph $\mathcal{G}_{i,j}$ is defined as a set of all the edges in the interval $[t_i, t_j]$, such that $\mathcal{E}_{ij} = \{e_k \mid t_i \leq t_k \leq t_j\}$. Any two nodes can interact multiple times throughout the time horizon; therefore, $\mathcal{G}$ is a multi-graph.

Our goal is to learn a model $f$ that maps the input graph to a representation space. The model is a pre-trainable encoder-decoder architecture, $f = (g_\theta, d_\gamma)$. The encoder $g_\theta$ maps a dynamic graph to node embeddings $\boldsymbol{H} \in \mathbb{R}^{N \times D^H}$; the decoder $d_\gamma$ performs a task-specific prediction given the embeddings. The model is parameterized by the encoder/decoder parameters $(\theta, \gamma)$. More concretely,

$$\boldsymbol{H} = g_\theta(\mathcal{G}), \qquad \boldsymbol{Z} = d_\gamma(\boldsymbol{H}; \bar{\mathcal{E}}), \tag{1}$$

where $\boldsymbol{Z} \in \mathbb{R}^{N \times D^Y}$ is the prediction of task-specific labels (e.g., edge prediction or source node classification labels) of all edges in $\bar{\mathcal{E}}$. The node embeddings $\boldsymbol{H}$ must capture the temporal and structural dynamics of each node such that the future can be accurately predicted from the past, e.g., future edge prediction given past edges. The main distinction of this design is that, unlike previous dynamic graph models (Rossi et al., 2020; Xu et al., 2020; Wang et al., 2021b), the encoder must produce embeddings independent of the downstream task specifications. This special trait can allow the model to be compatible with the SSL paradigm where an encoder is pre-trained separately and then fine-tuned together with a task-specific decoder to predict the labels.

To this end, we present a novel DyG2Vec framework, that can learn rich node embeddings at any timestamp $t$ independent of the downstream task. DyG2Vec is formulated as a two-stage framework. In the first stage, we use a non-contrastive SSL method to learn the model $f^{SSL} = (g_\theta, d_\psi)$ over various sampled dynamic sub-graphs with self-supervision. $d_\psi$ is an SSL decoder that is only used in the SSL pre-training stage. In the second stage, a task-specific decoder $d_\gamma$ is trained on top of the pre-trained encoder $g_\theta$ to compute the outputs for the downstream tasks, e.g., future edge prediction or dynamic node classification (Xu et al., 2020; Wang et al., 2021b).

We consider two example downstream tasks: future link prediction (FLP), and dynamic node classification (DNC). In each case, there is a prediction horizon of the next $K$ interactions. The test window for FLP starting at time $t_i$ is $\bar{\mathcal{E}} = \{(u_j, v_j, t_j, m_j) | j \in [i, i+K]\}$. This is augmented by a set of $K$ negative edges. Each negative edge $(u_j, v'_j, t_j, m_j)$ differs from its corresponding positive edge only in the destination node, $v'_j \neq v_j$, which is selected at random from all nodes. The FLP task is then binary classification for the test set of $2K$ edges. In the DNC task, a dynamic label is associated with each node that participates in an interaction. We are provided with $\{(u_j, t_j) | j \in [i, i+K]\}$, i.e., the source node and interaction time. The goal is to predict the source node labels for the next $K$ interactions. The performance metrics are detailed in Appendix A.4.

## 4 METHODOLOGY

We now introduce our novel dynamic graph learning framework DyG2Vec, which can achieve downstream task-agnostic representation. We first present the SSL pre-training approach with a non-contrastive loss function for dynamic graphs. We then introduce the novel window-based downstream training approach. Finally, we outline the encoder architecture.

Given the full dynamic graph $\mathcal{G}_{0,E}$, a set of intervals $I$ is generated by dividing the entire time-span $\{t_0, t_E\}$ into $M = \lceil E/S \rceil - 1$ intervals with stride $S$ and interval length $W$ (See Appendix A.4 for details). Let $B \subset I$ be the mini-batch of intervals. Given $B$, the sub-graph sampler $m(\mathcal{G}, B; W)$ constructs the mini-batch of input graphs: $\hat{\mathcal{G}} = \{\mathcal{G}_{i,j} \mid [i,j] \in B\}$. The corresponding mini-batch of target graphs is denoted by $\bar{\mathcal{G}} = \{\mathcal{G}_{j,j+K} \mid [i,j] \in B\}$. In principle, $\mathcal{G}_{i,j} \in \hat{\mathcal{G}}$ is an input (history) graph used to predict the target labels of the corresponding target (future) graph $\mathcal{G}_{j,j+K} \in \bar{\mathcal{G}}$. The parameter $W$ controls the size of the history while $K$ controls how far the model is predicting into

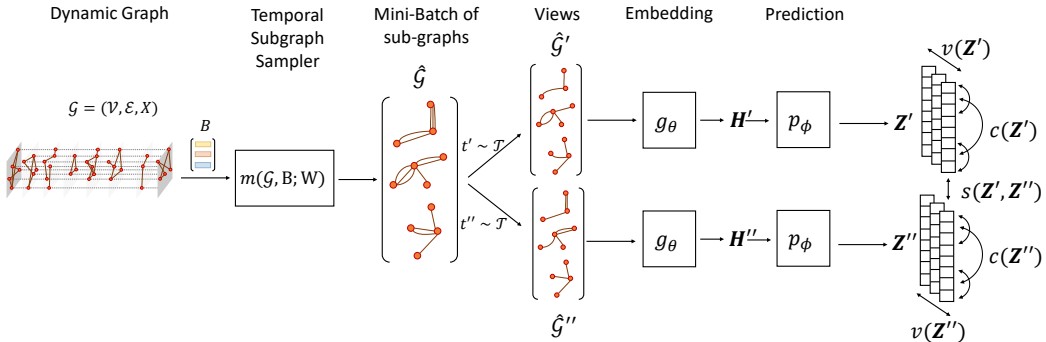

Figure 1: The joint embedding architecture for the non-contrastive SSL Framework. Each slice of the input dynamic graph contains edges arriving at the same continuous timestamp. $B$ is a batch of intervals of size $W$. $\hat{\mathcal{G}}$ is a batch of the corresponding input graphs of each interval.

the future. $S$ controls the stride between intervals. In practice, we set $S = K$ so that each edge is only predicted once in each epoch. Since $\bar{\mathcal{G}}$ is only provided for training in the downstream task, SSL pre-training only operates on $\hat{\mathcal{G}}$, as seen in Figure 2.

### 4.1 STAGE 1: PRE-TRAINING ON DYNAMIC GRAPHS WITH SELF-SUPERVISION

We formulate a joint-embedding architecture (Bromley et al., 1993) for DyG2Vec in which two views of a mini-batch of sub-graphs are generated through random transformations. The transformations are randomly sampled from a distribution defined by a distortion pipeline. The encoder maps the views to node embeddings which are processed by the predictor to generate node representations. We minimize an SSL objective (Eq. 2, described below) to optimize the model parameters end-to-end in the pre-training stage. See Figure 1 for an overall design of the SSL framework.

**Views**: The temporal distortion module generates two views of the input graphs $\hat{\mathcal{G}}' = t'(\hat{\mathcal{G}})$ and $\hat{\mathcal{G}}'' = t''(\hat{\mathcal{G}})$ where the transformations $t'$ and $t''$ are sampled from a distribution $\mathcal{T}$ over a pre-defined set of candidate graph transformations. In this work, we use edge dropout and edge feature masking (Thakoor et al., 2022) in the transformation pipeline. See Appendix A.4 for more details.

**Embedding**: The encoding model $g_\theta$ is an attention-based message-passing (AMP) neural network that produces node embeddings $\boldsymbol{H}'$ and $\boldsymbol{H}''$ for the views $\hat{\mathcal{G}}'$ and $\hat{\mathcal{G}}''$ of the input graphs $\hat{\mathcal{G}}_{i,j}$. We elaborate on the details of the encoder in Sec. 4.3.

**Prediction**: The decoding head $d_\gamma$ for our self-supervised learning design consists of a node-level predictor $p_\phi$ that outputs the final representations $\boldsymbol{Z}'$ and $\boldsymbol{Z}''$, where $\boldsymbol{Z} = p_\phi(\boldsymbol{H})$.

**SSL Objective**: In order to learn useful representations, we minimize the VICReg regularization-based SSL loss function from (Bardes et al., 2022):

$$\mathcal{L}^{SSL} = l(\boldsymbol{Z}', \boldsymbol{Z}'') = \lambda s(\boldsymbol{Z}', \boldsymbol{Z}'') + \mu[v(\boldsymbol{Z}') + v(\boldsymbol{Z}'')] + \nu[c(\boldsymbol{Z}') + c(\boldsymbol{Z}'')]. \qquad (2)$$

In this loss function, the weights $\lambda$, $\mu$, and $\nu$ control the emphasis placed on each of three regularization terms. The *invariance* term $s$ encourages representations of the two views to be similar. The *variance* term $v$ is included to prevent the well-known collapse problem (Jing et al., 2022). The covariance term $c$ promotes maximization of the information content of the representations. More details and complete expressions for $s$, $v$ and $c$ are provided in Appendix A.3.

Unlike previous regularization-based SSL approaches (Chen et al., 2020a; Bardes et al., 2022) in computer vision, we do not use a projector network because the embedding dimensions are relatively small in the graph domain. The full pre-training procedure is illustrated in Figure 1. Following the pre-training stage, we replace the SSL decoder with a task-specific downstream decoder $d_\psi$ that is trained on top of the *frozen* pre-trained encoder.

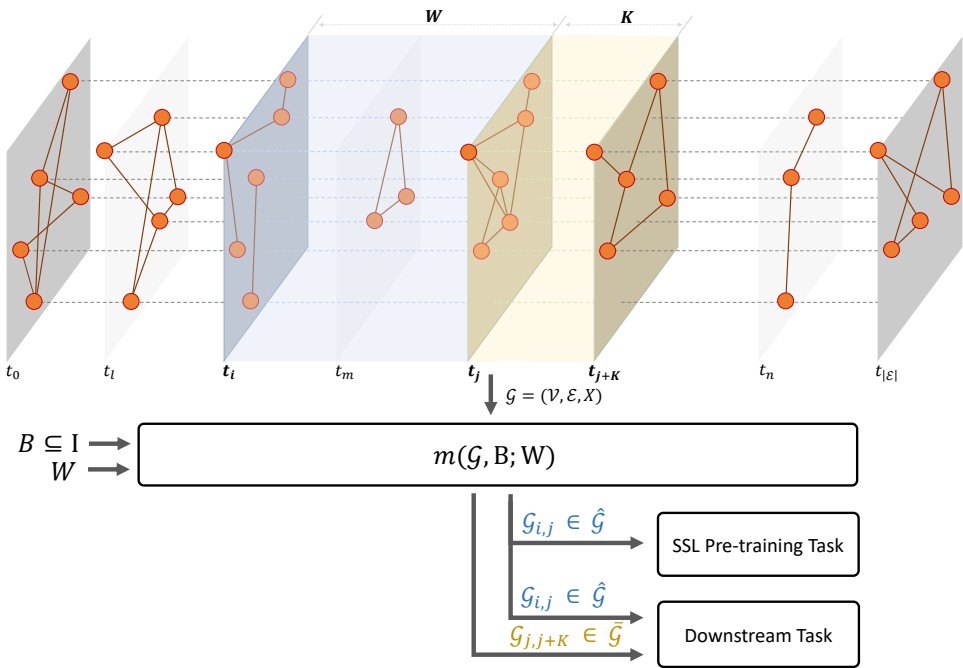

Figure 2: DyG2Vec Window Framework. Every slice of the dynamic graph $\mathcal{G}$ contains edges that arrived at the same continuous timestamp. The blue interval represents the history graph $\mathcal{G}_{i-W,i}$ that is encoded to make a prediction on the next $K$ edges (yellow interval). $B$ is a batch of intervals of size $W$ edges. $\hat{\mathcal{G}}$ is batch of input graphs. $\bar{\mathcal{G}}$ is batch of target graphs that is only used in the downstream stage.

## 4.2 STAGE 2: DYG2VEC DOWNSTREAM TRAINING

In the downstream training stage, the DyG2Vec model $f = (g_\theta, d_\psi)$ consists of the SSL pre-trained encoder $g_\theta$ and a task-specific decoder $d_\psi$, which is trained using a similar window-based training strategy. The model is trained to make predictions depending on the downstream tasks (e.g. link prediction or node classification) given the input and target graphs $\hat{\mathcal{G}}$ and $\bar{\mathcal{G}}$ as follows: $\boldsymbol{H} = g_\theta(\hat{\mathcal{G}})$ is the node embeddings returned by the encoder, and $\boldsymbol{Z} = d_\psi(\boldsymbol{H}; \bar{\mathcal{E}})$ is the prediction output of the decoder. Here $\bar{\mathcal{E}}$ is a set of (partial) edges for which predictions are requested from the decoder. The model parameters are optimized by training with a loss function $\mathcal{L}_D(\boldsymbol{Z}, \boldsymbol{O})$, where $\mathcal{L}_D$ is defined depending on the downstream task and $\boldsymbol{O}$ contains task-specific labels (See Section 3).

The window-based training strategy has several major advantages. First, the window acts as a regularizer by providing a natural inductive bias towards recent edges, which are often more predictive of the future. It also avoids costly time-based neighborhood sampling techniques (Wang et al., 2021b). Second, relying on a fixed window-size for message-passing allows for constant memory and computational complexity, which is well-suited to the practical *online streaming* data scenario. Third, unlike previous works (Xu et al., 2020; Wang et al., 2021b) which generate separate node embeddings for each target edge, a generic encoder allows us to use the same set of embeddings for any prediction. This dramatically reduces the training/inference overhead. Another advantage of this design is that it allows the model to forecast unseen edges relatively far into the future, in contrast to existing works (Xu et al., 2020; Rossi et al., 2020) that focus on predicting the next occurring edge.

## 4.3 DYG2VEC ENCODER ARCHITECTURE

Our encoder combines a self-attention mechanism for message-passing with the Time2Vec module (Kazemi et al., 2019) that provides relative time encoding. We also introduce a novel temporal edge encoding that efficiently captures the temporal structural relationship between nodes.

**Temporal Attention Embedding**: Given a dynamic graph $\mathcal{G}$, the encoder $g_\theta$ computes the embedding $\boldsymbol{h}_i^L$ of node $i$ through a series of $L$ multi-head attention (MHA) layers (Vaswani et al., 2017) that aggregate messages from its $L$-hop neighborhood (Xu et al., 2020; Velickovic et al., 2018).

Given a node embedding $\boldsymbol{h}_i^{l-1}$ at layer $l-1$, we uniformly sample $N$ 1-hop neighborhood interactions of node $i$, $\mathcal{N}(i) = \{e_p, \ldots, e_k\} \subseteq \mathcal{E}$. The embedding $\mathbf{h}_i^l$ at layer $l$ is calculated by:

$$\mathbf{h}_i^l = \mathbf{W}_1 \mathbf{h}_i^{l-1} + \text{MHA}^l(\mathbf{q}^l, \mathbf{K}^l, \mathbf{V}^l), \tag{3}$$

$$\mathbf{q}^l = \mathbf{h}_i^{l-1}, \tag{4}$$

$$\mathbf{K}^l = \mathbf{V}^l = [\Phi_p(t_p), \ldots, \Phi_k(t_k)]. \tag{5}$$

Here, $\mathbf{W}_1$ is a learnable mapping matrix, $\text{MHA}^l(\cdot)$ is a multi-head dot-product attention layer, and $\Phi_p(t_p)$ represents the edge feature vector of edge $e_p = (u_p, v_p, t_p, \boldsymbol{m}_p) \in \mathcal{N}(i)$ at time $t_p$:

$$\Phi_p(t_p) = [\boldsymbol{h}_{u_p}^{l-1} \mid\mid \boldsymbol{f}_p(t_p) \mid\mid \boldsymbol{m}_p], \tag{6}$$

$$\boldsymbol{f}_p(t_p) = \phi(\bar{t}_i - t_p) + \Theta_p(t_p), \tag{7}$$

$$\bar{t}_i = \max\{t_l \mid e_l \in \mathcal{N}(v_p).\}, \tag{8}$$

where $\mid\mid$ denotes concatenation and $\phi(.)$ is a learnable Time2Vec module that helps the model be aware of the relative timespan between a sampled interaction and the most recent interaction of node $v_p$ in the input graph. $\Theta_p(.)$ is a temporal edge encoding function, described in more detail below. In contrast to TGAT's recursive message passing procedure (Xu et al., 2020), the message passing in our encoder is 'flat': at every iteration, the same set of node embeddings is used to propagate messages to neighbors. Our encoder performs message passing once to generate a set of node embeddings $\boldsymbol{H}$ used for all target predictions on $\bar{\mathcal{G}}$.

**Temporal Edge Encoding**: Dynamic graphs often follow evolutionary patterns that reflect how nodes interact over time (Kovanen et al., 2011). For example, in social networks, two people who share many friends are likely to interact in the future. Therefore, we incorporate two simple yet effective temporal encoding methods that provide inductive biases to capture common structural and temporal evolutionary behaviour of dynamic graphs. The temporal edge encoding function is then:

$$\Theta_p(t_p) = \mathbf{W}_2[z_p(t_p) \mid\mid c_p(t_p)], \tag{9}$$

where we incorporate (i) *Temporal Degree Centrality* $z_p(t_p) \in \mathbb{R}^2$: the concatenated current degrees of nodes $u_p$ and $v_p$ at time $t_p$; and (ii) *Common Neighbors* $c_p(t_p) \in \mathbb{R}$: the number of common 1-hop neighbors between nodes $u_p$ and $v_p$ at time $t_p$.

By using the degree centrality as an edge feature, the model is able to learn any bias towards more frequent interactions with high-degree nodes. The number of common neighbors helps capture temporal motifs, and it is known to often have a strong positive correlation with the likelihood of a future interaction (Yao et al., 2016).

## 5 EXPERIMENTAL EVALUATION

### 5.1 EXPERIMENTAL SETUP

**Baselines**: We compare DyG2Vec to five state-of-the-art baseline models: DyRep (Trivedi et al., 2019), JODIE (Kumar et al., 2019), TGAT (Xu et al., 2020), TGN (Rossi et al., 2020), and CaW (Wang et al., 2021b). DyRep, JODIE, and TGN sequentially update node embeddings using an RNN. TGAT applies message passing via attention on a sampled temporal subgraph. CaW samples temporal random walks and learns temporal motifs by counting node occurrences in each walk.

**Downstream Tasks**: We evaluate all models on two temporal tasks: future link prediction (FLP), and dynamic node classification (DNC). In FLP, the goal is to predict the probability of future edges occurring given the source, destination, and timestamp. For each positive edge, we sample a negative edge that the model is trained to predict as negative. The DNC task involves predicting the label of the source node of a future interaction. Both tasks are trained using binary cross entropy loss. Contrary to prior works which only evaluate under the $K = 1$ setting, we evaluate all models under $K \in \{1, 200, 2000\}$. This is a more challenging evaluation scheme as it tests the forecasting capabilities of the models across multiple time horizons. See Appendix A.4 for details.

For the FLP task, we report both classification and recommendation metrics: Average Precision (AP), Mean Reciprocal-Rank (MRR), and Rec@10. For the DNC task, we report the area under the curve (AUC) metric due to the prevailing issue of class imbalance in dynamic graphs.

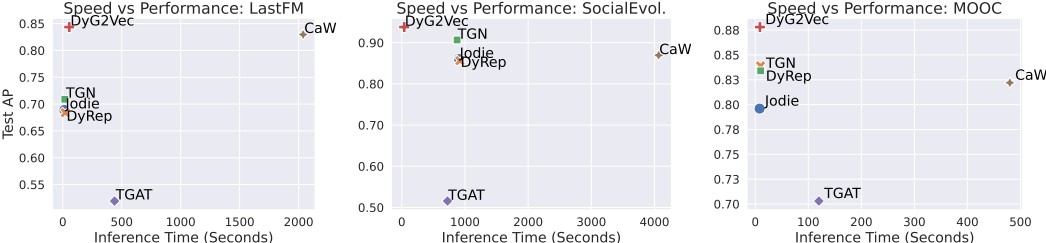

Figure 3: Transductive FLP Performance (Test AP for $K = 200$) vs Inference runtime (s) on 3 datasets. Inference time represents the time it takes to predict the whole test set.

Table 1: Transductive Future link Prediction Performance in AP (Mean $\pm$ Std). Avg. Rank reports the mean rank of DyG2Vec across all datasets. **Bold** font and underline font represent first- and second-best performance respectively.

| K | Model | Wikipedia | Reddit | MOOC | LastFM | Enron | UCI | SocialEvol. | Avg Rank |
|---|---|---|---|---|---|---|---|---|---|
| 1 | JODIE | $0.956 \pm 0.002$ | $0.979 \pm 0.001$ | $0.797 \pm 0.01$ | $0.691 \pm 0.010$ | $0.785 \pm 0.020$ | $0.869 \pm 0.010$ | $0.847 \pm 0.014$ | 4.71 |
| | DyRep | $0.955 \pm 0.004$ | $0.981 \pm 1e\text{-}4$ | $0.840 \pm 0.004$ | $0.683 \pm 0.033$ | $0.795 \pm 0.042$ | $0.524 \pm 0.076$ | $0.885 \pm 0.004$ | 4.85 |
| | TGAT | $0.968 \pm 0.001$ | $0.986 \pm 3e\text{-}4$ | $0.793 \pm 0.006$ | $0.633 \pm 0.002$ | $0.637 \pm 0.002$ | $0.835 \pm 0.003$ | $0.631 \pm 9e\text{-}4$ | 5.14 |
| | TGN | $\mathbf{0.986 \pm 0.001}$ | $0.985 \pm 0.001$ | $0.911 \pm 0.010$ | $0.743 \pm 0.030$ | $0.866 \pm 0.006$ | $0.843 \pm 0.090$ | $\mathbf{0.966 \pm 0.001}$ | 2.57 |
| | CaW | $0.976 \pm 0.007$ | $\mathbf{0.988 \pm 2e\text{-}2}$ | $\mathbf{0.940 \pm 0.014}$ | $\mathbf{0.903 \pm 1e\text{-}4}$ | $\mathbf{0.970 \pm 0.001}$ | $\mathbf{0.939 \pm 0.008}$ | $\underline{0.947 \pm 1e\text{-}4}$ | **1.57** |
| | **DyG2Vec** | $\underline{0.977 \pm 0.001}$ | $\underline{0.987 \pm 0.001}$ | $0.892 \pm 0.005$ | $\underline{0.843 \pm 0.007}$ | $\underline{0.911 \pm 0.006}$ | $\underline{0.953 \pm 0.002}$ | $0.939 \pm 0.001$ | 2.14 |
| 200 | JODIE | $0.956 \pm 0.003$ | $0.979 \pm 1e\text{-}4$ | $0.796 \pm 0.02$ | $0.689 \pm 0.018$ | $0.786 \pm 0.02$ | $0.870 \pm 0.009$ | $0.860 \pm 0.01$ | 4.42 |
| | DyRep | $0.954 \pm 0.003$ | $0.981 \pm 0.001$ | $\underline{0.839 \pm 0.005}$ | $0.684 \pm 0.034$ | $0.788 \pm 0.042$ | $0.522 \pm 0.081$ | $0.856 \pm 0.020$ | 4.71 |
| | TGAT | $0.958 \pm 0.002$ | $0.985 \pm 1e\text{-}4$ | $0.703 \pm 0.007$ | $0.519 \pm 0.002$ | $0.616 \pm 0.004$ | $0.727 \pm 0.010$ | $0.516 \pm 0.001$ | 5.14 |
| | TGN | $0.969 \pm 0.001$ | $0.985 \pm 0.001$ | $0.834 \pm 0.010$ | $0.709 \pm 0.030$ | $0.858 \pm 0.008$ | $0.762 \pm 0.110$ | $\underline{0.907 \pm 1e\text{-}4}$ | 3 |
| | CaW | $\mathbf{0.980 \pm 2e\text{-}4}$ | $\underline{0.985 \pm 0.001}$ | $0.822 \pm 0.001$ | $\underline{0.829 \pm 2e\text{-}4}$ | $\mathbf{0.929 \pm 0.001}$ | $0.856 \pm 0.001$ | $0.869 \pm 0.001$ | 2.28 |
| | **DyG2Vec** | $0.968 \pm 0.001$ | $\mathbf{0.986 \pm 0.001}$ | $\mathbf{0.878 \pm 0.004}$ | $\mathbf{0.844 \pm 0.005}$ | $\underline{0.894 \pm 0.006}$ | $\mathbf{0.918 \pm 0.002}$ | $\mathbf{0.938 \pm 0.001}$ | **1.42** |
| 2000 | JODIE | $0.924 \pm 0.004$ | $0.960 \pm 0.001$ | $0.740 \pm 0.020$ | $0.614 \pm 0.010$ | $0.729 \pm 0.010$ | $0.676 \pm 0.040$ | $0.691 \pm 0.023$ | 4.92 |
| | DyRep | $\underline{0.966 \pm 0.002}$ | $0.966 \pm 0.002$ | $0.776 \pm 0.005$ | $0.590 \pm 0.032$ | $0.729 \pm 0.042$ | $0.511 \pm 0.072$ | $0.704 \pm 0.044$ | 4.07 |
| | TGAT | $0.951 \pm 0.001$ | $\underline{0.985 \pm 1e\text{-}4}$ | $0.686 \pm 0.007$ | $0.513 \pm 0.003$ | $0.618 \pm 0.004$ | $0.702 \pm 0.010$ | $0.509 \pm 0.001$ | 5 |
| | TGN | $0.955 \pm 0.001$ | $0.984 \pm 0.001$ | $0.760 \pm 0.010$ | $0.689 \pm 0.030$ | $0.842 \pm 0.010$ | $0.746 \pm 0.020$ | $0.839 \pm 0.007$ | 3.28 |
| | CaW | $\mathbf{0.971 \pm 3e\text{-}4}$ | $\mathbf{0.984 \pm 2e\text{-}4}$ | $0.732 \pm 0.001$ | $\underline{0.811 \pm 1e\text{-}4}$ | $\mathbf{0.929 \pm 0.001}$ | $0.766 \pm 0.003$ | $0.847 \pm 0.001$ | 2 |
| | **DyG2Vec** | $0.956 \pm 0.001$ | $0.984 \pm 0.002$ | $\mathbf{0.798 \pm 0.002}$ | $\mathbf{0.829 \pm 0.004}$ | $\underline{0.879 \pm 0.006}$ | $\mathbf{0.866 \pm 0.002}$ | $\mathbf{0.908 \pm 0.001}$ | **1.71** |

**Datasets**: We use 7 real-world datasets: Wikipedia, Reddit, MOOC, and LastFM (Kumar et al., 2019); SocialEvolution, Enron, and UCI (Wang et al., 2021b). These datasets span a wide range in terms of number of nodes and interactions, time range, and repetition ratio. We perform the same 70%-15%-15% chronological split for all datasets as in (Wang et al., 2021b). The datasets are split differently under two settings: Transductive and Inductive. More details can be found in Appendix A.1. The code and datasets will be publicly available upon publication.

**Training Protocols and Hyper-parameters**: We train and evaluate the models under three different settings commonly used in vision SSL works (Grill et al., 2020; Bardes et al., 2022). In the supervised setting, DyG2Vec is initialized with random parameters and trained directly on the downstream tasks and compared to all supervised baselines. In the self-supervised setting, the encoder is pre-trained using our SSL framework, and the performance is measured under two evaluation protocols: Linear and Semi-supervised Probing. In the linear evaluation setting, the decoder is trained on top of the frozen encoder and compared to the supervised counterpart. In the semi-supervised evaluation setting, the decoder is trained on top of the frozen pre-trained encoder on a random portion of the dataset (i.e., a fraction of the intervals $I$). The DyG2Vec encoder performs $L = 3$ layers of message passing. We sample $N = 20$ temporal neighbors at each hop as in Xu et al. (2020). Other hyperparameters are discussed in Appendix A.5.

For the DNC task, following prior work Rossi et al. (2020), the decoder is trained on top of the frozen encoder that is pre-trained on the future link prediction task unless otherwise explicitly stated.

## 5.2 EXPERIMENTAL RESULTS

**Future Link Prediction**: We report the tranductive test AP scores for future link prediction in Table 1. Unlike previous work which focuses on $K = 1$, we evaluate all models under $K \in \{1, 200, 2000\}$ to test their medium- and longer-term forecasting capabilities. Unsurprisingly, all methods degrade as $K$ increases. Our model significantly outperforms all sequential and message-passing baselines on 5/7 of the datasets for $K > 1$ and is on par with SoTA (CaW) for $K = 1$. The gap is particularly large on the UCI and SocialEvol. datasets for $K = 2000$, where DyG2Vec outperforms the second-

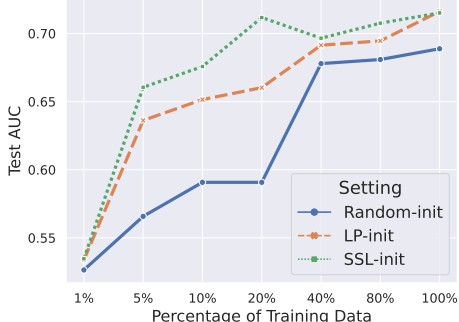

Figure 5: Semi-Supervised Learning on Dynamic Node Classification. For each setting, DyG2Vec was trained on a varying random portion of the training data.

Table 2: Transductive Dynamic Node Classification Performance in AUC (Mean $\pm$ Std) for $K \in \{1, 200, 2000\}$.

| Model | Wikipedia | Reddit | MOOC |
|---|---|---|---|
| TGAT | $0.800 \pm 0.01$ | $0.664 \pm 0.009$ | $0.673 \pm 0.006$ |
| JODIE | $\underline{0.843 \pm 0.003}$ | $0.566 \pm 0.016$ | $0.672 \pm 0.002$ |
| Dyrep | $\mathbf{0.873 \pm 0.002}$ | $0.633 \pm 0.008$ | $0.661 \pm 0.012$ |
| TGN | $0.828 \pm 0.004$ | $\underline{0.655 \pm 0.009}$ | $\underline{0.674 \pm 0.007}$ |
| **DyG2Vec** | $0.833 \pm 0.008$ | $\mathbf{0.744 \pm 0.01}$ | $\mathbf{0.716 \pm 0.006}$ |
| TGAT | $0.819 \pm 0.014$ | $0.654 \pm 0.011$ | $0.664 \pm 0.004$ |
| JODIE | $0.843 \pm 0.003$ | $0.566 \pm 0.016$ | $\underline{0.672 \pm 0.002}$ |
| Dyrep | $\underline{0.872 \pm 0.002}$ | $0.633 \pm 0.009$ | $0.664 \pm 0.011$ |
| TGN | $0.834 \pm 0.003$ | $\underline{0.655 \pm 0.011}$ | $0.641 \pm 0.009$ |
| **DyG2Vec** | $\mathbf{0.927 \pm 0.004}$ | $\mathbf{0.745 \pm 0.006}$ | $\mathbf{0.683 \pm 0.002}$ |
| TGAT | $0.828 \pm 0.017$ | $0.650 \pm 0.020$ | $\underline{0.667 \pm 0.010}$ |
| JODIE | $0.867 \pm 0.002$ | $0.554 \pm 0.019$ | $0.652 \pm 0.008$ |
| Dyrep | $\underline{0.873 \pm 0.002}$ | $0.597 \pm 0.012$ | $\mathbf{0.692 \pm 0.002}$ |
| TGN | $0.840 \pm 0.003$ | $0.646 \pm 0.009$ | $0.638 \pm 0.030$ |
| **DyG2Vec** | $\mathbf{0.915 \pm 0.003}$ | $\mathbf{0.743 \pm 0.007}$ | $0.609 \pm 0.002$ |

best method (CaW) by over $10\%$ and $6\%$ respectively. Interestingly, while SocialEvol. is the largest dataset with $\sim 2M$ edges, our model is able to achieve this performance while only using the last 1000 edges (See Table 9) to predict any future edge. This further cements the findings by Xu et al. (2020) that capturing recent interactions may be more important for certain tasks. Our window-based framework offers a good trade-off between capturing recent interactions and recurrent patterns which both have a major influence on future interactions. Appendix A.2 contains results in the Inductive setting which show that DyG2Vec is competitive with CaW for 4/7 of the datasets while using a small fraction of the computation (See Figure 3).

**Dynamic Node classification**: We evaluate DyG2Vec on 3 datasets for node classification where the labels indicate whether a user will be banned from editing/posting after an interaction. This task is challenging both due to its dynamic nature (i.e., nodes can change labels) and the high class imbalance (only 217 of 157K interactions result in a ban). We measure performance using the AUC metric to deal with the class imbalance. Table 2 shows that DyG2Vec outperforms all baselines on 7/9 of the tasks for different $K$. Interestingly, the performance of all methods does not always drop as $K$ increases. This could be explained by the fact that depending on slightly out-of-date history can help make less noisy and more consistent predictions; thus, increasing performance.

**Training/Inference Speed**: Relying on a fixed window of history to produce task-agnostic node embeddings gives DyG2Vec a significant advantage in speed and memory. Figure 3 shows the performance and runtime per epoch of all methods on the three large datasets: LastFM, SocialEvolution and MOOC, with $K = 200$. DyG2Vec is many orders of magnitude faster than CaW due to the latter's expensive random walk sampling procedure. RNN-based methods such as TGN have a good runtime on LastFM and MOOC; however, they are significantly slower on SocialEvol. which has a small number of nodes (74) but a large number of interactions ($\sim 2M$). This suggests that memory-based methods are slower for settings where a node's memory is updated frequently. Furthermore, while TGAT has a similar AMP encoder, DyG2Vec improves the efficiency and performance significantly. This reveals the significance of the window-based mechanism and the encoder architecture. Overall, DyG2Vec presents the best trade-off between speed and performance.

**SSL for Future Link Prediction**:
Table 3 reports the transductive AP results for DyG2Vec under 3 different settings. Namely, we compare a random frozen encoder (Random-init) and an SSL pre-trained encoder (SSL-init) with the supervised baseline. The results reveal that our SSL pre-training learns informative node embeddings that are almost on par with the fully supervised baseline. This supports the capability of the

Table 3: Linear probing AP results (Mean $\pm$ Std) on Transductive Future Link Prediction for $K \in \{1, 200, 2000\}$

| Setting | UCI | Enron | MOOC | LastFM |
|---|---|---|---|---|
| Random-init | $0.927 \pm 0.001$ | $0.848 \pm 0.004$ | $0.785 \pm 0.007$ | $0.784 \pm 0.001$ |
| Supervised | $0.953 \pm 0.002$ | $0.911 \pm 0.006$ | $0.892 \pm 0.005$ | $0.843 \pm 0.007$ |
| SSL-init | $0.945 \pm 0.001$ | $0.902 \pm 0.002$ | $0.861 \pm 0.003$ | $0.798 \pm 0.01$ |
| Random-init | $0.904 \pm 0.001$ | $0.839 \pm 0.005$ | $0.787 \pm 0.002$ | $0.784 \pm 0.002$ |
| Supervised | $0.918 \pm 0.002$ | $0.894 \pm 0.006$ | $0.878 \pm 0.004$ | $0.844 \pm 0.005$ |
| SSL-init | $0.915 \pm 0.001$ | $0.887 \pm 0.003$ | $0.859 \pm 0.002$ | $0.792 \pm 0.01$ |
| Random-init | $0.851 \pm 0.001$ | $0.832 \pm 0.004$ | $0.742 \pm 0.003$ | $0.776 \pm 0.001$ |
| Supervised | $0.866 \pm 0.002$ | $0.879 \pm 0.006$ | $0.798 \pm 0.002$ | $0.829 \pm 0.004$ |
| SSL-init | $0.863 \pm 0.002$ | $0.876 \pm 0.007$ | $0.789 \pm 0.003$ | $0.785 \pm 0.01$ |

non-contrastive methods to learn generic representations across unlabelled large-scale dynamic

graphs, which is in line with the findings for other data modalities (Bardes et al., 2022). The Random-init baseline is surprisingly good, as observed by recent works (Thakoor et al., 2022), but is outperformed by the SSL pre-trained encoder.

**Semi-supervised Learning on Dynamic Node Classification**: The DNC task is challenging due to its highly imbalanced labels. Previous works alleviate this issue by pre-training the encoder on future link prediction. In Figure 5, we show that SSL is a more effective pre-training strategy for dynamic graphs than FLP, particularly in the low-label data regime where each model is trained on a portion of the target intervals $I$. This further cements the findings that reconstruction-based tasks such as link prediction overemphasize proximity which can be limiting for some downstream tasks (Velickovic et al., 2019; You et al., 2020).

## 5.3 ABLATION STUDIES

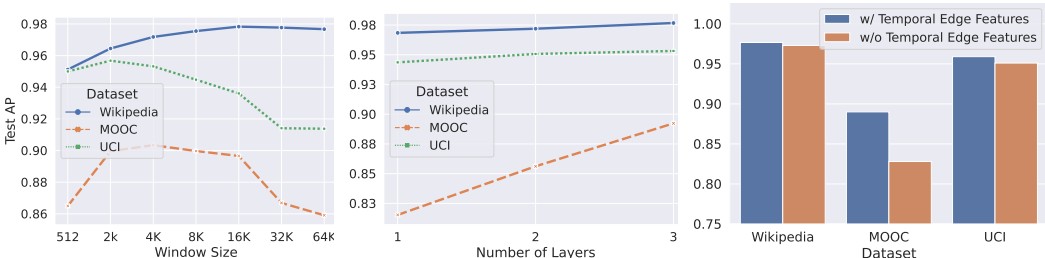

Figure 6: Ablation studies on 3 datasets for the FLP transductive task.

We perform a detailed study on different instances of our framework with 3 datasets. All ablation results are reported in Figure 6.

**Window Size**: We observe that each dataset has its own optimal window size $W$ due to the inherently different recurring temporal patterns. As observed by Xu et al. (2020), recent and/or recurrent interactions are often the most predictive of future interactions. Therefore, datasets with long range dependencies favor larger window sizes to capture the recurrent patterns while some datasets benefit from an increased bias towards recent interactions. Our window-based framework coupled with uniform neighbor sampling strikes a balance between the two. Moreover, increasing the window size to 64K for UCI, which is effectively full history as it has 60K edges in total, results in a 4% drop in performance. This shows that the fixed window size also contributes to the performance as it helps limit irrelevant information that is not highly predictive of future interactions.

**Number of Layers:** Most datasets benefit from more embedding layers, and some (e.g. MOOC) more than others. This suggests that these datasets contain higher order temporal correlations among nodes that must be learned using long-range message passing. Overall, the results show that one can choose to sacrifice some performance to further improve the speed of DyG2Vec by decreasing the window size and the number of layers.

**Temporal Edge Features**: The results show a significant decrease in performance for MOOC when temporal edge features are removed (i.e. 1-5% drop). This indicates that such temporal edge features provide useful multi-hop information about the evolution of the dynamic graph (Yao et al., 2016).

## 6 CONCLUSION

In this paper, we introduce DyG2Vec, a novel window-based encoder-decoder model for dynamic graphs. We present an efficient attention-based message-passing model that utilizes hierarchical multi-head attention modules to encode node embeddings across time. Furthermore, we present a joint-embedding architecture for dynamic graphs in which two views of temporal sub-graphs are encoded to minimize a non-contrastive loss function. We evaluate the SSL pre-training of DyG2Vec under both linear and semi-supervised protocols and demonstrate the effectiveness of such pre-training on benchmark datasets. Our window-based architecture allows for efficient message-passing and robust forecasting abilities. We aim to further explore ways to improve the capacity of the dynamic graph models to learn long-range dependencies. Additionally, other SSL paradigms such as masked auto-encoders are worthwhile future explorations for dynamic graphs given their success on sequential tasks.

## 7 ETHICS STATEMENT

Dynamic graph neural network techniques have been commonly used for prediction tasks in social networks and recommender systems. Our techniques, as an efficient and effective variant of dynamic graph representation technique, can be used in those scenarios to further improve the model performance. However, having such an ability is a double-edged sword. On one hand, it can be beneficial to greatly improve user experience. On the other hand, there may be some concerns about the potential use of the model to exploit data relating to user behaviour and thus invade privacy. Overall, our paper does not include content which has immediate ethical concerns.

## 8 REPRODUCIBILITY STATEMENT

We describe the details of the benchmarks we used in Appendix A.1. We also include the full implementation details in Appendix A.4, including our detailed model design for decoder, negative sampler, distortion pipeline for view generations and etc. Besides, we provide a justification for our choice of hyperparameters in Appendix A.5. Additionally, we elaborate on the baselines we consider in our paper as well as their hyper-parameter details in Appendix A.6. We believe with the code open-sourced upon acceptance and the detailed description of our model and the baselines provided in the paper, our work is fully reproducible.

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

# A APPENDIX

## A.1 DATASETS

Table 4: Dynamic Graph Datasets. **% Repetitive Edges**: percentage of edges which appear more than once in the dynamic graph. The labels column specifies whether the dataset contains dynamic node labels or not.

| | # Nodes | # Edges | # Unique Edges | Edge Features | Labels | Bipartite | % Repetitive Edges |
|---|---|---|---|---|---|---|---|
| Reddit | 11,000 | 672,447 | 78,516 | ✓ | ✓ | ✓ | 54% |
| Wikipedia | 9,227 | 157,474 | 18,257 | ✓ | ✓ | ✓ | 48% |
| MOOC | 7,047 | 411,749 | 178,443 | | ✓ | ✓ | 53% |
| LastFM | 2,000 | 1,293,103 | 154,993 | | | ✓ | 68% |
| UCI | 1899 | 59,835 | 20,296 | | | ✓ | 62% |
| Enron | 184 | 125,235 | 3,125 | | | | 92% |
| SocialEvolution | 74 | 2,099,519 | 4,486 | | | | 97% |

We describe several real-world dynamic graph datasets which we train on:

**Reddit** (Kumar et al., 2019): A dataset tracking active users posting in subreddits. Data is represented as a bipartite graph with nodes being the users or subreddit communities. An edge represents a user posting on a subreddit. Each user's post is mapped to an embedding vector which is used as an edge feature. A Dynamic label indicates whether a user $u$ is banned from posting after an interaction (post) at time $t$.

**Wikipedia** (Kumar et al., 2019): A dataset tracking user edits on Wikipedia pages. The data is also represented as a bipartite graph involving interactions (edits) between users and Wikipedia pages. Each user edit is mapped to an embedding vector which is treated as an edge feature. A dynamic label indicates whether a user $u$ is banned from posting after an interaction (edit) at time $t$.

**MOOC** (Kumar et al., 2019): This dataset tracks actions performed by students on the MOOC online course platform. Nodes represent students or items (i.e., videos, questions, etc.). A dynamic label indicates whether a student $u$ drops out after performing an action at time $t$.

**LastFM** (Kumar et al., 2019): This dataset tracks songs that users listen to throughout one month. Nodes represent users or songs. Dynamic labels are not present.

**UCI** (Wang et al., 2021b): A dataset that records online posts made by university students on a forum.

**Enron** (Wang et al., 2021b): A dataset that records email communication between employees in a company over several years.

**SocialEvolve** (Wang et al., 2021b): A dataset tracking the evolving physical proximity between students in a dormitory over a year.

**Dataset Splitting**: As mentioned earlier, we follow the setup of (Wang et al., 2021b) to perform 70%-15%-15% chronological split for each dataset. The datasets are split differently under two settings: Transductive and Inductive. Under the transductive setting, a dataset is split normally by time, i.e., the model is trained on the first 70% of links and tested on the rest. In the inductive setting, we strive to test the model's prediction performance on edges with unseen nodes. Therefore, following (Wang et al., 2021b), we randomly assign $10\%$ of the nodes to the validation and test sets and remove any interactions involving them in the training set. Additionally, to ensure an inductive setting, we remove any interactions not involving these nodes from the test set.

## A.2 ADDITIONAL RESULTS

### A.2.1 INDUCTIVE SETTING AND RANKING METRICS

Table 5 reports the future link prediction results in the inductive setting. DyG2Vec is generally competitive with CaW while using 50-100× less inference and training runtime (See 3 and 7 respectively). Tables 6 and 7 report the future link prediction ranking performance for $K = 1$ on the transductive and inductive tasks, respectively. While ranking metrics should provide more fine-grained analysis, the results are consistent with the AP results (See Tables 1 and 5).

Table 5: Inductive Future link Prediction Performance in AP (Mean ± Std) for $K \in \{1, 200, 2000\}$. Avg. Rank reports the mean rank of DyG2Vec across all datasets. **Bold** font and underline font represent first- and second-best performance respectively.

| K | Model | Wikipedia | Reddit | MOOC | LastFM | Enron | UCI | SocialEvolve | Avg Rank |
|---|---|---|---|---|---|---|---|---|---|
| | JODIE | $0.891 \pm 0.014$ | $0.865 \pm 0.021$ | $0.707 \pm 0.029$ | $0.865 \pm 0.03$ | $0.747 \pm 0.041$ | $0.753 \pm 0.011$ | $0.791 \pm 0.031$ | 4.85 |
| | DyRep | $0.890 \pm 0.002$ | $0.921 \pm 0.003$ | $0.723 \pm 0.009$ | $0.869 \pm 0.015$ | $0.666 \pm 0.059$ | $0.437 \pm 0.021$ | $0.904 \pm 3e\text{-}4$ | 4.71 |
| | TGAT | $0.954 \pm 0.001$ | $0.979 \pm 0.001$ | $0.805 \pm 0.006$ | $0.644 \pm 0.002$ | $0.693 \pm 0.004$ | $0.820 \pm 0.005$ | $0.632 \pm 0.005$ | 4.14 |
| 1 | TGN | $0.974 \pm 0.001$ | $0.954 \pm 0.002$ | $0.855 \pm 0.014$ | $0.789 \pm 0.050$ | $0.746 \pm 0.013$ | $0.791 \pm 0.057$ | $0.904 \pm 0.023$ | 3.42 |
| | CaW | **$0.977 \pm 0.006$** | **$0.984 \pm 2e\text{-}4$** | **$0.933 \pm 0.014$** | $0.890 \pm 0.001$ | **$0.962 \pm 0.001$** | **$0.931 \pm 0.002$** | **$0.950 \pm 1e\text{-}4$** | **1.28** |
| | DyG2Vec | $0.964 \pm 0.002$ | $0.969 \pm 0.001$ | $0.799 \pm 0.010$ | **$0.910 \pm 0.010$** | $0.827 \pm 0.023$ | $0.945 \pm 0.005$ | $0.850 \pm 0.038$ | 2.57 |
| | JODIE | $0.891 \pm 0.012$ | $0.865 \pm 0.021$ | $0.704 \pm 0.028$ | $0.866 \pm 0.031$ | $0.748 \pm 0.040$ | $0.762 \pm 0.010$ | $0.824 \pm 0.025$ | 4.14 |
| | DyRep | $0.884 \pm 0.002$ | $0.919 \pm 0.003$ | $0.723 \pm 0.019$ | $0.864 \pm 0.016$ | $0.648 \pm 0.071$ | $0.431 \pm 0.021$ | $0.86 \pm 0.024$ | 4.57 |
| | TGAT | $0.932 \pm 0.002$ | **$0.977 \pm 0.001$** | $0.724 \pm 0.006$ | $0.532 \pm 0.002$ | $0.5993 \pm 0.009$ | $0.698 \pm 0.009$ | $0.5178 \pm 0.003$ | 4.14 |
| 200 | TGN | $0.923 \pm 0.005$ | $0.948 \pm 0.003$ | $0.745 \pm 0.016$ | $0.769 \pm 0.063$ | $0.701 \pm 0.019$ | $0.683 \pm 0.083$ | $0.601 \pm 0.028$ | 4.28 |
| | CaW | **$0.978 \pm 0.001$** | $0.975 \pm 0.001$ | **$0.812 \pm 0.009$** | $0.787 \pm 0.002$ | **$0.905 \pm 0.004$** | $0.819 \pm 0.003$ | **$0.871 \pm 0.001$** | **1.71** |
| | DyG2Vec | $0.925 \pm 0.004$ | $0.963 \pm 0.001$ | $0.774 \pm 0.009$ | **$0.901 \pm 0.020$** | $0.773 \pm 0.047$ | **$0.880 \pm 0.007$** | $0.836 \pm 0.038$ | 2.14 |
| | JODIE | $0.718 \pm 0.029$ | $0.712 \pm 0.028$ | $0.588 \pm 0.013$ | $0.755 \pm 0.034$ | $0.423 \pm 0.032$ | $0.727 \pm 0.045$ | $0.687 \pm 0.110$ | 4.57 |
| | DyRep | $0.757 \pm 0.006$ | $0.842 \pm 0.006$ | $0.599 \pm 0.010$ | $0.682 \pm 0.032$ | $0.402 \pm 0.012$ | $0.483 \pm 0.011$ | $0.687 \pm 0.049$ | 5.00 |
| | TGAT | $0.912 \pm 0.002$ | **$0.976 \pm 0.001$** | $0.699 \pm 0.004$ | $0.528 \pm 0.001$ | $0.578 \pm 0.007$ | $0.660 \pm 0.008$ | $0.511 \pm 0.003$ | 3.14 |
| 2000 | TGN | $0.857 \pm 0.003$ | $0.895 \pm 0.005$ | $0.655 \pm 0.018$ | $0.737 \pm 0.068$ | $0.478 \pm 0.024$ | $0.531 \pm 0.039$ | $0.483 \pm 0.002$ | 4.43 |
| | CaW | **$0.976 \pm 0.001$** | $0.975 \pm 0.001$ | **$0.742 \pm 0.011$** | $0.759 \pm 3e\text{-}4$ | **$0.865 \pm 0.003$** | **$0.737 \pm 0.007$** | **$0.833 \pm 0.001$** | **1.28** |
| | DyG2Vec | $0.860 \pm 0.007$ | $0.942 \pm 0.003$ | $0.71 \pm 0.015$ | **$0.890 \pm 0.02$** | $0.498 \pm 0.033$ | $0.533 \pm 0.007$ | $0.789 \pm 0.035$ | 2.57 |

Table 6: Tranductive Setting. Ranking metrics, Recall@10 and Mean Reciprocal Rank (MRR), for future link prediction $K = 1$. Avg. Rank reports the mean rank of DyG2Vec across all datasets

| Metric | Model | Wikipedia | Reddit | MOOC | LastFM | Enron | UCI | SocialEvol. | Avg Rank |
|---|---|---|---|---|---|---|---|---|---|
| | JODIE | $0.686 \pm 0.010$ | $0.800 \pm 0.003$ | $0.443 \pm 0.040$ | $0.131 \pm 0.030$ | $0.262 \pm 0.050$ | $0.258 \pm 0.040$ | $0.728 \pm 0.026$ | 5.00 |
| | DyRep | $0.667 \pm 0.019$ | $0.815 \pm 0.005$ | $0.562 \pm 0.013$ | $0.139 \pm 0.049$ | $0.288 \pm 0.079$ | $0.010 \pm 0.014$ | $0.786 \pm 0.010$ | 4.57 |
| Recall@10 | TGAT | $0.717 \pm 0.010$ | $0.840 \pm 0.003$ | $0.297 \pm 0.013$ | $0.023 \pm 0.001$ | $0.106 \pm 0.002$ | $0.154 \pm 0.004$ | $0.147 \pm 0.001$ | 5.00 |
| | TGN | $0.865 \pm 0.003$ | **$0.861 \pm 0.004$** | $0.751 \pm 0.050$ | $0.208 \pm 0.040$ | $0.470 \pm 0.030$ | $0.339 \pm 0.100$ | **$0.947 \pm 0.003$** | 2.14 |
| | CaW | **$0.888 \pm 0.001$** | $0.833 \pm 0.001$ | $0.706 \pm 0.012$ | **$0.411 \pm 0.002$** | **$0.720 \pm 0.002$** | **$0.72 \pm 0.008$** | $0.752 \pm 2e\text{-}4$ | 2.14 |
| | DyG2Vec | $0.797 \pm 0.005$ | $0.837 \pm 0.002$ | **$0.753 \pm 0.009$** | $0.365 \pm 0.007$ | $0.696 \pm 0.008$ | $0.670 \pm 0.010$ | $0.922 \pm 0.003$ | 2.14 |
| | JODIE | $0.371 \pm 0.040$ | $0.499 \pm 0.030$ | $0.183 \pm 0.010$ | $0.057 \pm 0.010$ | $0.108 \pm 0.020$ | $0.101 \pm 0.010$ | $0.301 \pm 0.019$ | 5.28 |
| | DyRep | $0.422 \pm 0.015$ | $0.558 \pm 0.009$ | $0.224 \pm 0.007$ | $0.066 \pm 0.022$ | $0.118 \pm 0.03$ | $0.010 \pm 0.006$ | $0.353 \pm 0.010$ | 4.42 |
| MRR | TGAT | $0.464 \pm 0.016$ | $0.609 \pm 0.004$ | $0.131 \pm 0.006$ | $0.015 \pm 3e\text{-}4$ | $0.053 \pm 0.001$ | $0.111 \pm 0.002$ | $0.071 \pm 0.0005$ | 4.85 |
| | TGN | $0.684 \pm 0.030$ | $0.594 \pm 0.030$ | **$0.364 \pm 0.030$** | $0.100 \pm 0.020$ | $0.206 \pm 0.010$ | $0.242 \pm 0.070$ | **$0.666 \pm 0.017$** | 2.42 |
| | CaW | **$0.717 \pm 0.001$** | $0.603 \pm 0.002$ | $0.314 \pm 0.010$ | **$0.191 \pm 0.001$** | **$0.315 \pm 0.003$** | **$0.601 \pm 0.004$** | $0.323 \pm 2e\text{-}4$ | **1.85** |
| | DyG2Vec | $0.596 \pm 0.009$ | **$0.636 \pm 0.011$** | $0.272 \pm 0.004$ | $0.164 \pm 0.002$ | $0.254 \pm 0.009$ | $0.360 \pm 0.010$ | $0.404 \pm 0.002$ | 2.14 |

## A.2.2 EFFECT OF K DURING TRAINING

One of the advantages of the DyG2Vec framework is that, unlike prior work, it is trained to forecast with $K > 1$. In fact, the parameter $K$ can be set during training based on how much long-range range forecasting is favored. In our experiments, we found that training with $K = 200$ achieves a good balance between long and short-range forecasting capabilities. Moreover, the limited history of $W$ edges forces the model to be more inductive as it is predicting based on limited long-range historical information. To better understand this, we trained both DyG2Vec and CaW under $K = \{1, 200\}$ and evaluate on $K = \{1, 200, 2000\}$. Table 8 shows that training DyG2Vec with short-range prediction ($K = 1$) improves performance to be on par with CaW on $K = 1$ and outperform it for $K > 1$. However, as expected, this comes at a cost of $\sim 2\%$ drop for long-range forecasting ($K > 1$) when compared to DyG2Vec trained with $K = 200$. On the other hand, CaW's performance drops significantly when trained with $K = 200$ (i.e. over 10% drop for UCI and MOOC). We believe this is due to the sampling bias $\alpha$ which may be incorrectly favoring recent edges over edges that occurred further in the past but can help for long-range forecasting. Unfortunately, we were unable to address this by re-tuning $\alpha$. An interesting direction for future research would be to study training settings under which all models can have improved forecasting abilities.

Table 7: Inductive Setting. Ranking metrics, Recall@10 and Mean Reciprocal Rank (MRR), for future link prediction $K = 1$. Avg. Rank reports the mean rank of DyG2Vec across all datasets

| Metric | Model | Wikipedia | Reddit | MOOC | LastFM | Enron | UCI | SocialEvol. | Avg Rank |
|---|---|---|---|---|---|---|---|---|---|
| | JODIE | $0.450 \pm 0.068$ | $0.275 \pm 0.048$ | $0.291 \pm 0.048$ | $0.475 \pm 0.065$ | $0.231 \pm 0.099$ | $0.086 \pm 0.031$ | $0.658 \pm 0.082$ | 5.00 |
| | DyRep | $0.487 \pm 0.009$ | $0.534 \pm 0.016$ | $0.327 \pm 0.019$ | $0.479 \pm 0.057$ | $0.144 \pm 0.073$ | $0.003 \pm 0.001$ | $0.833 \pm 0.006$ | 4.14 |
| Recall@10 | TGAT | $0.634 \pm 0.002$ | **$0.792 \pm 0.005$** | $0.326 \pm 0.011$ | $0.026 \pm 4e\text{-}4$ | $0.103 \pm 0.006$ | $0.175 \pm 0.007$ | $0.147 \pm 0.002$ | 4.57 |
| | TGN | $0.776 \pm 0.012$ | $0.681 \pm 0.009$ | $0.585 \pm 0.025$ | $0.451 \pm 0.034$ | $0.234 \pm 0.016$ | $0.261 \pm 0.043$ | **$0.860 \pm 0.021$** | 2.71 |
| | CaW | **$0.894 \pm 0.002$** | $0.770 \pm 0.005$ | **$0.688 \pm 0.016$** | $0.327 \pm 0.0023$ | **$0.687 \pm 0.010$** | **$0.709 \pm 0.003$** | $0.773 \pm 0.0001$ | 2.14 |
| | DyG2Vec | $0.742 \pm 0.017$ | $0.734 \pm 0.004$ | $0.543 \pm 0.030$ | **$0.535 \pm 0.020$** | $0.269 \pm 0.017$ | $0.616 \pm 0.020$ | $0.774 \pm 0.057$ | 2.42 |
| | JODIE | $0.192 \pm 0.048$ | $0.098 \pm 0.020$ | $0.121 \pm 0.017$ | $0.186 \pm 0.028$ | $0.084 \pm 0.029$ | $0.043 \pm 0.009$ | $0.301 \pm 0.067$ | 4.85 |
| | DyRep | $0.322 \pm 0.015$ | $0.301 \pm 0.016$ | $0.130 \pm 0.007$ | $0.179 \pm 0.039$ | $0.061 \pm 0.025$ | $0.006 \pm 0.001$ | **$0.427 \pm 0.012$** | 4.28 |
| MRR | TGAT | $0.404 \pm 0.009$ | **$0.528 \pm 0.009$** | $0.146 \pm 0.004$ | $0.016 \pm 2e\text{-}4$ | $0.052 \pm 0.003$ | $0.133 \pm 0.008$ | $0.071 \pm 3e\text{-}4$ | 4.42 |
| | TGN | $0.606 \pm 0.005$ | $0.440 \pm 0.009$ | **$0.295 \pm 0.019$** | $0.127 \pm 0.015$ | $0.115 \pm 0.011$ | $0.186 \pm 0.025$ | $0.423 \pm 0.075$ | 2.71 |
| | CaW | **$0.712 \pm 0.004$** | $0.521 \pm 0.002$ | $0.292 \pm 0.013$ | $0.158 \pm 0.001$ | **$0.298 \pm 0.005$** | **$0.584 \pm 0.004$** | $0.351 \pm 0.001$ | **2.28** |
| | DyG2Vec | $0.564 \pm 0.025$ | $0.523 \pm 0.030$ | $0.198 \pm 0.010$ | **$0.204 \pm 0.008$** | $0.105 \pm 0.006$ | $0.290 \pm 0.019$ | $0.362 \pm 0.035$ | 2.42 |

Table 8: Future link prediction test AP Results with different K during training. Both CaW and DyG2Vec were run with optimal hyperparameters for each dataset.

| Training $K$ | Model | MOOC | | | UCI | | | Enron | | |
|---|---|---|---|---|---|---|---|---|---|---|
| | | 1 | 200 | 2000 | 1 | 200 | 2000 | 1 | 200 | 2000 |
| $K = 1$ | DyG2Vec | 93.0 | 84.3 | 74.5 | 97.2 | 89.6 | 84.03 | 96.2 | 91.9 | 90.3 |
| | CaW | 94.0 | 82.2 | 73.2 | 93.9 | 85.6 | 76.6 | 97.0 | 92.9 | 92.9 |
| $K = 200$ | DyG2Vec | 89.2 | 87.8 | 79.8 | 95.3 | 91.8 | 86.6 | 91.1 | 89.4 | 87.9 |
| | CaW | 81.9 | 61.2 | 60.8 | 84.2 | 63.6 | 58.0 | 91.8 | 78.6 | 78.0 |

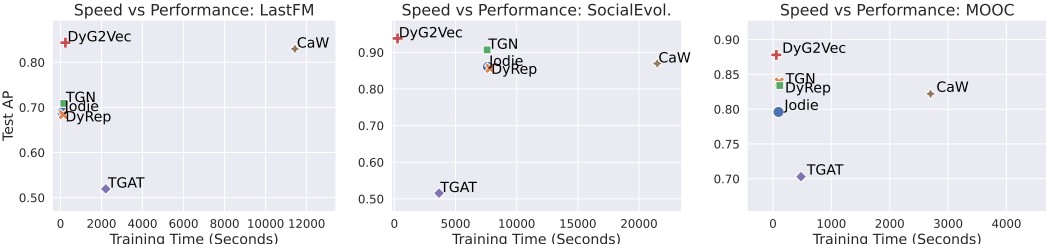

Figure 7: Transductive FLP performance (Test AP for $K = 200$) vs Training Time (s) on 3 datasets. Training time represents the time it takes to train on the whole training set.

### A.2.3 RUNTIME AND COMPUTATIONAL COMPLEXITY

Figure 7 shows the training time per method for 3 datasets. DyG2Vec is orders-of-magnitude faster than CaW and is on par with memory-based methods in terms of speed.

The main runtime overhead lies in how each of the baselines processes the input graph to predict a batch of $K$ target edges. CaW samples $M$ $L$-hop random walks for each target edge. This is followed by an expensive set-based anonymization scheme. To achieve good performance, CaW can require relatively long walks (e.g., for Enron, $L = 5$). On the other hand, memory-based methods and TGAT sample a different $L$-hop subgraph for each target edge. DyG2Vec samples a single $L$-hop subgraph within a constant window size $W$ for all target edges.

Thus, assuming we use sparse operations in Pytorch Geometric (Fey & Lenssen, 2019) for message-passing, the encoding computational complexities are: DyG2Vec = $O(LW)$; CaW = $O(LMN_sK)$ and TGN and variants = $O(KLN_s)$. Here, $N_s$ represents the maximum number of sampled nodes in an $L$-hop subgraph and $K$ is the number of target edges to predict. We can see that the main difference is the factor $M$ and the fact that this sampling is done for each of the $K$ target edges. The factor $N_s$ comes from the complexity of message passing at each hop (assuming sparse operations). Note that DyG2Vec is limited to $O(W)$ nodes so it does not have this factor.

### A.3 PRELIMINARY: VICREG

We outline the details of the VICReg (Bardes et al., 2022) method used in our SSL pre-training stage. Given $z^{'} \in \mathbb{R}^d$ and $z^{''} \in \mathbb{R}^d$, the representations of two random views of an object (e.g., image) generated through random distortions, the objective of non-contrastive SSL is two-fold. First, the output representation of one view should be maximally informative of the input representation of the view. Second, the representation of one view should be maximally predictable from the representation of the other view. These two aspects are formulated by VICReg (Bardes et al., 2022) where a combination of 3 loss terms (i.e. Variance, Covariance, Invariance) is minimized to learn useful representations while also avoiding the well-known problem of collapse (Jing et al., 2022) in the mapping space. More concretely, let $\boldsymbol{Z}^{'} = [\boldsymbol{z}_1', \dots, \boldsymbol{z}_n']$ and $\boldsymbol{Z}^{''} = [\boldsymbol{z}_1'', \dots, \boldsymbol{z}_n'']$ be the batches composed of $n$ representations of dimension $d$.

**Variance term**: The variance regularization term $v$ is the mean over the representation dimension of the hinge function on the standard deviation of the representations along the batch dimension: $v(\boldsymbol{Z}) = \frac{1}{D} \sum_{j=1}^{D} \max(0, \gamma - S(\boldsymbol{Z}_{:,j}, \epsilon))$. Here $\boldsymbol{Z}_{:,j}$ is the column $j$ of matrix $\boldsymbol{Z}$. $S$ is the regu-

larized standard deviation defined by $S(\boldsymbol{z}, \epsilon) = \sqrt{\text{Var}(\boldsymbol{z}) + \epsilon}$, $\gamma$ is a constant value set to 1 in our experiments, and $\epsilon$ is a small scalar that helps to prevent numerical instability. This term avoids dimensional collapse by maximizing the volume of the distribution of the mapped views in all dimensions. In other words, it prevents the well-known trivial solution where the representations of the two views of a sample collapse to the same representation (Jing et al., 2022).

**Covariance term**: The covariance regularization terms $C$ decorrelates different dimensions of the representations and prevents them from encoding similar information. The covariance matrix of $\boldsymbol{Z}$ is $C(\boldsymbol{Z}) = \frac{1}{n} \sum_{i=1}^{n} (\boldsymbol{z}_i - \bar{\boldsymbol{z}})(\boldsymbol{z}_i - \bar{\boldsymbol{z}})^T$ where $\bar{\boldsymbol{z}} = \frac{1}{N} \sum_{n}^{i=1} \boldsymbol{z}_i$. The covariance regularization term $c$ is then defined as the sum of the squared off-diagonal coefficients of the covariance matrix as follows $c(\boldsymbol{Z}) = \frac{1}{d} \sum_{i \neq j} [C(\boldsymbol{Z})]_{i,j}^2$ where $[C(\boldsymbol{Z})]_{i,j}^2$ is the element at row $i$ and column $j$ of the matrix $C(\boldsymbol{Z})$. Both the variance and covariance terms helps to maximize the information encoded by the model in the representation space.

**Invariance criterion**: The invariance criterion $s$ between $\boldsymbol{Z}'$ and $\boldsymbol{Z}''$ is defined as the mean squared Euclidean distance between the representation vectors in the two views $s(\boldsymbol{Z}', \boldsymbol{Z}'') = \frac{1}{n} \sum_i \|\boldsymbol{z}_i' - \boldsymbol{z}_i''\|_2^2$. The invariance term encourages the parametric mapping to ensure that the views of an object remain close in the latent space.

Finally, the SSL loss function $\mathcal{L}^{SSL}$ over a batch of representations is a weighted average of the invariance, variance, and covariance terms:

$$\mathcal{L}^{SSL} = l(\boldsymbol{Z}', \boldsymbol{Z}'') = \lambda s(\boldsymbol{Z}', \boldsymbol{Z}'') + \mu[v(\boldsymbol{Z}') + v(\boldsymbol{Z}'')] + \nu[c(\boldsymbol{Z}') + c(\boldsymbol{Z}'')] \tag{10}$$

In our experiments, we set $\lambda = \mu = 25$ and $\nu = 1$, following Bardes et al. (2022).

### A.4 IMPLEMENTATION DETAILS

We train our model using the Pytorch framework (Paszke et al., 2019). The dynamic graph data and GNN encoder architecture are implemented using Pytorch Geometric (Fey & Lenssen, 2019). The ReLU activation function is used for all models. The code and datasets are publicly available [2].

**Window-based framework**: As mentioned in Section 4, the full dynamic graph $G_{0,E}$ is divided into a set of intervals $I$ that is generated by dividing the entire time-span into $M = \lceil E/S \rceil - 1$ intervals with stride $S$ and interval length $W$:

$$I = \left\{ \left[ \max(0, jS - W), \min(jS, E) \right) \mid j \in \{1, 2, \dots, M\} \right\}. \tag{11}$$

Here, $W$ defines the number of edges in an interval and $S$ defines the stride. Note that we include all intervals up to but not including $[E - W, E)$ so that the target interval contains at least one edge.

**Negative Sampling**: For future link prediction, we sample an equal number of negative interactions and positive (target) interactions. Negative interactions are sampled by sampling a random negative destination node from the set of all possible nodes. For the Recall@10 and MRR metrics, we sample 500 random negative destinations for each positive target edge.

**Decoder Architecture**:

Denote by $t^{max}$ the timestamp of the latest interaction, within the provided history, incident to node $u$. For future link prediction, to predict a target interaction $(u, v, t)$, our decoder maps the sum of the two node embeddings of $u$ and $v$ and a time embedding of $t - t^{max}$ to an edge probability. Following Xu et al. (2020), the FLP decoder is a 2-layer MLP.

For dynamic node classification, to predict the label of node $u$ for interaction $(u, v, t)$, the decoder maps the source node embedding and time embedding of $t - t^{max}$ to class probabilities. Following Xu et al. (2020), the DNC decoder is a 3-layer MLP with a dropout layer with $p = 0.1$.

The time embedding is calculated using a trainable Time2Vec module (Kazemi et al., 2019). The time embedding allows the decoder to be time-aware; hence, possibly output different predictions for the same nodes/edges at different timestamps.

---

[2]https://github.com/anon/anon.git

For SSL pre-training, the predictor $p_\phi$ is a simple 2-layer MLP that maps node embeddings $\boldsymbol{H}$ to node representations $\boldsymbol{Z}$.

**Distortion Pipeline**: Following static graph SSL methods such as Thakoor et al. (2022), we use edge dropout and edge feature dropout as distortions. Both distortions are applied with dropout probability $p_d = 0.3$ which we have found to work best in a validation experiment exploring the values $p_d \in \{0.1, 0.15, 0.2, 0.3\}$. The edge feature dropout is applied on the temporal edge encodings introduced in Section 4.3, i.e., $z_p(t_p)$ and $c_p(t_p)$. More advanced temporal distortions could be explored such as time distortions or edge shuffling, but we leave those for future work.

## A.5 HYPER-PARAMETERS

**Hyper-parameters**: We use a constant learning rate of 0.0001 for all datasets and tasks. DyG2Vec is trained for 100 epochs for both downstream and SSL pre-training. The model from the last epoch of pre-training is used for downstream training. For downstream evaluation, we pick the model with the best validation AP performance. Overall, we found that DyG2Vec converges within $\sim 50$ epochs.

For downstream training, the window size $W$ is tuned on the validation set from the range $\{1K, 4K, 8K, 12K, 32K, 65K\}$. The best window sizes per dataset can be found in Table 9. The target window size $K$ is fixed to 200 during training. The stride is always fixed to be equal to $K$, so that each edge is only predicted once. One could augment the dataset by changing $S$ but we leave this for future work. The batch size is always set to 1; hence, we only predict one target interval of size $K$ at a time. However, the model could be sped up by increasing batch size at the cost of higher memory.

During SSL pre-training, we use a constant window size of 32K with stride 200. The DyG2Vec encoder hyperparameters can be found in Table 10.

Table 9: Optimal Window size $W$ for downstream training.

| Dataset | Window Size $W$ |
|---|---|
| Wikipedia | 65K |
| Reddit | 65K |
| MOOC | 12K |
| LastFM | 32K |
| UCI | 4K |
| Enron | 8K |
| SocialEvolution | 1K |

Following previous work (Rossi et al., 2020; Xu et al., 2020), all dynamic node classification training experiments are performed with L2-decay parameter $\lambda = 0.00001$ to alleviate over-fitting.

## A.6 BASELINES

**Baselines:** Following prior work (Rossi et al., 2020; Xu et al., 2020), all baselines are trained with a constant learning rate of 0.0001 using the Adam optimizer (Kingma & Ba, 2014) on batch-size 200 for a total of 50 epochs. The early stopping strategy is used to stop training if validation AP does not improve for 5 epochs. For JODIE (Kumar et al., 2019), DyRep (Trivedi et al., 2019), and TGN (Rossi et al., 2020), we use the general framework[3] implemented by Rossi et al. (2020). The node memory dimension is set to 172. Other encoder hyperparameters are specified in Table 10.

For TGAT[4], we use the default hyperparameters of 2 layer neighbor sampling with 20 neighbors sampled at each hop. Other hyperparameters are specified in Table 10. For the CaW[5] method, we tune the time decay parameter $\alpha \in \{0.01, 0.1, 0.3, 0.5, 1, 2, 4, 10\} \times 10^{-6}$, and length of the walks $m \in \{2, 3, 4, 5\}$ on the validation set. The optimal hyperparameters for each dataset are specified in Table 11. The number of heads for the walking-based attention is fixed to 8.

---

[3]https://github.com/twitter-research/tgn
[4]https://github.com/StatsDLMathsRecomSys/Inductive-representation-learning-on-temporal-graphs
[5]https://github.com/snap-stanford/CAW

Table 10: Hyperparameters for DyG2Vec, RNN-based methods (JODIE, DyRep, and TGN), and TGAT.

| Param | Value |
|---|---|
| Node Embedding Dim | 100 |
| Time Embedding Dim | 100 |
| # Attention Heads | 2 |
| # Sampled Neighbors | 20 |
| Dropout | 0.1 |

Table 11: Hyperparameters for CaW.

| Dataset | Time Decay $\alpha$ | Walk Length $m$ |
|---|---|---|
| Wikipedia | 4e-6 | 4 |
| Reddit | 1e-8 | 3 |
| MOOC | 1e-4 | 3 |
| LastFM | 1e-6 | 3 |
| UCI | 1e-5 | 2 |
| Enron | 1e-6 | 5 |
| SocialEvolution | 3e-5 | 3 |

### A.7 COMPUTING INFRASTRUCTURE

All experiments were done on a Ubuntu 20.4 server with 72 Intel(R) Xeon(R) Gold 6140 CPU @ 2.30GHz cores and a RAM of size 755 Gb. We use a NVIDIA Tesla V100-PCIE-32GB GPU.

### A.8 ADDITIONAL RELATED WORK

**Self-supervised learning for dynamic graphs:** Most adaptations of SSL for dynamic graphs have focused on improving downstream task performance via auxiliary losses rather than learning general pre-trained models. Jiang et al. (2021) adapt a sub-graph contrastive learning method (Jiao et al., 2020) where a node representation is contrasted in both structure and time. That is, for each node in the graph, a GNN encoder is trained to contrast its real temporal subgraph to its fake temporal sub-graph . This is done by constructing a positive sample, a structural negative sample and a temporal negative sample. The positive sample is a time-weighted subgraph representation. The margin triplet loss is proposed to maximize the mutual information with the positive sample while maximizing distance with the structural and temporal negative samples. Experiments on downstream link prediction task under the freeze setting show improvement over baselines. However, their approach comes with several shortcomings. First, initial node features are computed as one-hot encodings which makes the method not suitable for the inductive scenario (i.e. predicting on new nodes). Second, the use of contrastive learning method is known to result in high memory and computation due to negative sampling (Thakoor et al., 2022). This makes the method less desirable for large-scale graphs. Third, they do not include results on other downstream tasks (e.g. dynamic node classification). Lastly, they do not compare to the SoTA CaW method (Wang et al., 2021b).

Cong et al. (2022) propose the dynamic graph transformer (DGT) which is a transformer-based graph encoder for *discrete-time dynamic graphs*. DGT is composed of two-tower networks that embed the temporal evolution and topological information of the input graph. Moreover, a temporal-union graph structure is proposed to efficiently summarize the temporal evolution into one graph. DGT is trained to encode the temporal-union graph using two complementary self-supervised pretext tasks. Namely, temporal reconstruction and multi-view contrasting. The first aims to reconstruct a snapshot given the past and present similar to how language models are trained. On the other hand, the latter is trained via non-contrastive learning on two views with randomly masked nodes. All together, DGT outperforms SOTA discrete-time baselines on several datasets for link prediction tasks. While they operate in a different domain, an interesting direction for future work would be to adapt their pre-training strategy for continuous-time dynamic graphs.

Table 12: Downstream Freeze test AP Results (after pre-training) for K=1. DDGCL pre-training and downstream training were run with default parameters described in the work. DyG2Vec was run with parameters described in A.5.

| Model | MOOC | Enron | UCI | LastFM |
|---|---|---|---|---|
| DyG2Vec | **86.1** | **90.1** | **94.5** | **79.8** |
| DDGCL | 84.3 | 83.0 | 85.3 | 78.8 |

Tian et al. (2021) adapt the TGAT encoder with a self-supervised contrastive framework across time. That is, they propose an extension to the classic contrastive learning paradigm by contrasting two nearby temporal views of the same node using a time-dependent similarity metric. Moreover, a de-basied contrastive loss is utilized to correct the typical negative sampling bias in contrastive learning. Experiments on the fine-tune and mutli-task learning settings show that the simple TGAT encoder can be significantly improved on both future link prediction and dynamic node classification. Nonetheless, their approach comes with several shortcomings. First, it is built on the TGAT encoder which, as seen in Tables 1 and 5, is a weak encoder; particularly, for large datasets. Second, experiments for the FLP task are limited to the Reddit and Wikipedia datasets which are relatively easy. Lastly, the authors do not experiment under the standard settings in graph SSL literature such as the freeze and semi-supervised settings. Table 12 shows the results for downstream future link prediction under the freeze setting. The results show up to 10% gap compared to DyG2Vec, particularly for datasets where the TGAT encoder under-performs (e.g. Enron, UCI).

**More encoders for temporal graphs**: Souza et al. (2022) is a concurrent work that establishes a series of theoretical results on temporal graph encoders. Their analysis exposes several weakness of both memory-based methods (e.g. TGN) and walk-based methods (e.g. CaW). Given these insights, they propose PINT, a memory-based method that leverages injective message-passing and novel relative positional encodings. The relative positional encodings count how many temporal walks of a given length exist between two nodes. Experiments show significant improvement over SoTA baselines on the link prediction task. An interesting direction for future research would be to evaluate the expressive power of DyG2Vec compared to baselines using their theoretical framework (e.g. temporal WL test).

Wang et al. (2021a) adapt the vanilla transformer architecture to dynamic graphs by designing a two-stream encoder that extracts temporal and structural information from the temporal neighborhoods associated with any two interaction nodes. Rather than treating link prediction as a binary classification task, the authors leverage a contrastive learning strategy that maximizes the mutual information between the representations of future interaction nodes. Experiments show improved performance on future link prediction due to the more robust contrastive training strategy. Nonetheless, the paper does not compare to the SoTA CaW method (Wang et al., 2021b). Moreover, experiments are limited to the future link prediction task.

