# OpenReview forum: "DyG2Vec: Representation Learning for Dynamic Graphs With Self-supervision"
_ICLR.cc/2023/Conference — Submitted to ICLR 2023_

### Official Review · Reviewer_yV1s · 2022-10-25

**Confidence:** 4
**Correctness:** 3
**Technical Novelty And Significance:** 2
**Empirical Novelty And Significance:** 2
**Recommendation:** 3

**Clarity, Quality, Novelty And Reproducibility:**

Clarity: Presentation can be largely improved.
Quality: Extensive experiments but lack of deeper analyses.
Novelty: Propose an interesting SSL mechanism, but similar to related works and it is not well explained why the mechanism works for dynamic graphs.
Reproducibility: Datasets, settings, protocals, hyperparameters, and infrastructures are included in the appendix.

**Strength And Weaknesses:**

[+] A new SSL learning method for dynamic graphs by using a window-based mechanism.

[+] Experiments on benchmark datasets and two downstream tasks show that the method has stronger prediction ability and less inference time than SOTA baselines.

[+] The paper also introduces SSL pretraining and finetuning protocals for dynamic graphs.

[-] Limited novelty and unclear motivation. a) the proposed window-based mechanism seems straight-forward adaption from ssl learning for static graphs, and the view generators for ssl, i.e. edge dropping and edge feature masking in this paper, are commonly adopted in graph ssl literature. It is expected to design specific ssl methods by utilizing the dynamics which can not be considered in static graphs. b) It is not well explained why the window-based mechanism acts as a good ssl method for dynamic graphs. c) The motivation of SSL for dynamic graphs is also not well-explained. The authors state that "task labels are often scarce, costly to obtain, and highly imbalanced for large dynamic graphs", but it seems not supported in dynamic graph literature. The authors are expected to give support for the statement in detail. Moreover, in link prediction, which is common downstream task in dynamic graph literature, edges serve as training task labels (historic edges) and also testing task labels (future edges), and it does not support the statement "task labels are often scarce".

[-] Presentation can be largely improved.

[-] Lack of deeper analyses. For example, it is not well-explained why the proposed method has better forecasting capability in larger horizon.

[-] Missing important related works. There are a few related works about self-supervised dynamic graph learning methods, e.g [1-4].

[1] Tian, Sheng, et al. "Self-supervised Representation Learning on Dynamic Graphs." Proceedings of the 30th ACM International Conference on Information & Knowledge Management. 2021.
[2] Cong, Weilin, et al. "Dynamic Graph Representation Learning via Graph Transformer Networks." arXiv preprint arXiv:2111.10447 (2021).
[3] Wang, Lu, et al. "Tcl: Transformer-based dynamic graph modelling via contrastive learning." arXiv preprint arXiv:2105.07944 (2021).
[4] Jiang, Linpu, Ke-Jia Chen, and Jingqiang Chen. "Self-Supervised Dynamic Graph Representation Learning via Temporal Subgraph Contrast." arXiv preprint arXiv:2112.08733 (2021).

(Minor)
1. typo in Figure 6.

**Summary Of The Paper:**

This paper proposes a SSL-compatible, efficient dynamic graph model named DyG2Vec, which uses a window-based mechanism to generate task-agnostic node embeddings. Experiments on benchmark datasets and two downstream tasks show that the method has stronger prediction ability and less inference time than SOTA baselines. The paper also introduces SSL pretraining and finetuning protocals for dynamic graphs.


**Summary Of The Review:**

The paper proposes a new SSL learning method for dynamic graphs by using a window-based mechanism and extensive experiments show that the method has stronger prediction ability and less inference time than SOTA baselines. However, the reviewer concerns that 1) the motivation and presentation is not clear 2) the technical novelty is limited 3) importance related works are missing 4) deeper analyses of the proposed method is missing.

---

> ### Author Response · Authors · 2022-11-15
> **Author Response Part 1**
>
> Thank you for your valuable time and review! Please find our response below.
>
> > 4.1 the proposed window-based mechanism seems straight-forward adaption from ssl learning for static graphs, and the view generators for ssl, i.e. edge dropping and edge feature masking in this paper, are commonly adopted in graph ssl literature. It is expected to design specific ssl methods by utilizing the dynamics which can not be considered in static graphs
>
> While it is true that the distortions are adapted from the static graph SSL literature, we want to emphasize that applying SSL methods in such temporal domains is not trivial as dynamic graphs can involve heavy distribution shifts. For example, new nodes arrive and others depart, and these arrival patterns occur at different timescales. Therefore, we believe that it is important to understand how these SSL algorithms perform in temporal domains and under what conditions pre-training is helpful (e.g. low-label regime). Our development of methods and experimental results thus constitute a valuable resource for the community.
>
> As far as we know, we are the first to experiment with non-contrastive methods on dynamic graphs and analyze learned embeddings in the freeze and semi-supervised settings. Our results show promising improvements for low-label regime (2-5\% improvement in Fig. 6) and near supervised performance in the freeze setting (Table 3 in paper). We are not aware of previous work that provide such insights. Moreover, our window-based framework uses a fraction of the computational and memory cost of other baselines making it a good candidate for large-scale pre-training. Therefore, we believe the extensive analysis the we have provided across a variety of datasets and tasks opens a promising door for future explorations on learning on dynamic graphs purely from observation.
>
>
> Moreover, in effort of designing SSL algorithms that take advantage of the temporal domain, we have explored distortions that operate on the dynamic aspect but have not found any significant performance gains. Table 1 below shows the downstream freeze results given pre-training with different distortions. We introduce the temporal masking distortion where we randomly drop out a patch of consecutive interactions within a window. Since close interactions can be often be highly correlated with each other, this makes the SSL task harder for the model as it must extrapolate to force the node representations of the 2 views to be similar; thus, learning more rich representations. Nonetheless, we do not see any improvements of such distortions over the simple edge and feature dropout.
>
>
> Table 1: Test AP results on Downstream link prediction task with $K = 1$. Both results are under the freeze setting. ”Setting” column defines the distortions used during pre-training. Here, temporal masking represents removing a random patch of 1K consecutive edges with probability
> 0.5.
>
> |                    Setting                    |   UCI  |  Enron |
> |:---------------------------------------------:|:------:|:------:|
> | Edge Dropout + Edge Feature Dropout (Default) | $94.5$ | $90.2$ |
> |        Edge Dropout + Temporal Masking        | $93.6$ | $88.9$ |
>
>
> > 4.2 It is not well explained why the window-based mechanism acts as a good ssl method for dynamic graphs
>
> We thank the reviewer for raising this point. The main point of window-based SSL pre-training is for the model to learn fine-grained temporal patterns within the graph. Applying the SSL loss on the full dynamic graph makes it hard for the model to discover  temporal motifs that develop over small time-spans, especially for large-scale dynamic graphs (e.g. recommendation systems). In Table 2 below, we show the downstream freeze results given pre-training with $W=32K$ (default) versus $W=80K$. Note that pre-training with $W=80K$ is the same as pre-training on the full dynamic graph because the window size encompasses the full graph (the training set of Enron has only 70K interactions). Overall, we observe that DyG2Vec favors pre-training on (smaller) windows of the dynamic graph. This can be explained by the fact that the window-based training serves as a natural mechanism to control the information flow into the model; therefore, avoiding the common over-squashing and over-smoothing phenomena for large graphs.
>
> Table 2: Test AP results on Downstream link prediction task with $K=1$. Both results are under the freeze setting. Here, $W$ represents the window size during pre-training. The window size during downstream task is fixed to 8K.
>
> |   Setting   |  Enron |
> |:-----------:|:------:|
> | SSL $W=32k$ | $90.2$ |
> | SSL $W=80K$ | $88.8$ |
>
> > 4.3 Presentation can be largely improved.
>
> We have revised Section 4 in the main paper and added more informative figure captions. We are happy to further improve our presentation. We would appreciate it if you can provide more details on what needs to be improved.

---

> ### Author Response · Authors · 2022-11-15
> **Author Response Part 2**
>
> > 4.4 The motivation of SSL for dynamic graphs is also not well-explained. The authors state that "task labels are often scarce, costly to obtain, and highly imbalanced for large dynamic graphs", but it seems not supported in dynamic graph literature. The authors are expected to give support for the statement in detail. Moreover, in link prediction, which is common downstream task in dynamic graph literature, edges serve as training task labels (historic edges) and also testing task labels (future edges), and it does not support the statement "task labels are often scarce"
>
> We note that link prediction is a common downstream task in graph SSL literature [1] used to benchmark different SSL algorithms. While the task itself does not require manual labels, it serves as an experimental framework for evaluating how much structural and temporal information is preserved in the SSL embeddings.
>
> The practical advantage of SSL lies in tasks with scarce or unbalanced labels such as anomaly detection (i.e. dynamic node classification). For example, for the dynamic node classification datasets Reddit (0.05\%), Wikipedia (0.1\%), and MOOC (1\%), fewer than 1\% of interactions result in a ban (positive label); therefore, directly training on this task is challenging. In general, the lack of balanced and rich labels is also common for other dynamic domains such videos where per-frame labeling is difficult to attain. Prior works [3] in such domains found that the temporal aspect can serve as a rich supervisory signal to learn without labels. In dynamic graphs, previous work alleviates data imbalance by pre-training on the future link prediction task. However, prior work [2] in the static domain shows that this results in models failing to learn general node embeddings and instead focusing on proximity. Therefore, we show in Fig. 5 how SSL outperforms link prediction as a pre-training method, particularly in the low-label regime (e.g. only 5-20\% of labels available), where DyG2Vec outperforms link prediction pre-training by up to 5\%.
>
> [1] Liu, Y., Pan, S., Jin, M., Zhou, C., Xia, F., and Yu, P.S. 2022. Graph Self-Supervised Learning: A Survey. arXiv preprint arXiv:2103.00111.
>
> [2] Velickovic, P., Fedus, W., Hamilton, W.L., Lio, P., Bengio, Y., and Hjelm, R.D. 2019. Deep Graph Infomax. In Proceedings for the International Conference on Learning Representations.
>
> [3] Tong, Z., Song, Y., Wang, J., and Wang, L. 2022. VideoMAE: Masked Autoencoders are Data-Efficient Learners for Self-Supervised Video Pre-Training. Advances in Neural Information Processing Systems.

---

> ### Author Response · Authors · 2022-11-15
> **Author Response Part 3**
>
> > 4.5 Lack of deeper analyses. For example, it is not well-explained why the proposed method has better forecasting capability in larger horizon.
>
> We thank the reviewer for raising this concern. One of the advantages of DyG2Vec is that, unlike prior work, it is trained to forecast with $K>1$. In our experiments, we found that training with $K=200$ gives a good trade-off between long and short-range forecasting capabilities. Moreover, the limited history of $W$ edges forces the model to be more inductive as it is predicting based on limited long-range historical information. To better understand this, we trained both DyG2Vec and CaW under $K=\{1, 200\}$ and evaluate on $K=\{1, 200, 2000\}$. Table 3 below shows that training DyG2Vec with short-range prediction ($K=1$) improves performance to be on par with CaW on $K=1$ and outperform it for $K > 1$. However, as expected, this comes at a cost of $\sim 2\%$ drop for long-range forecasting ($K > 1$) when compared to DyG2Vec trained with $K = 200$. On the other hand, CaW's performance drops significantly when trained with $K=200$ (i.e. over 10\% drop for UCI and MOOC). We believe this is due to the sampling bias $\alpha$ which may be incorrectly favoring recent edges over far edges that can help for long-range forecasting. Unfortunately, we were unable to fix this by re-tuning $\alpha$. An interesting direction for future research would be to study training settings under which all models can have improved forecasting abilities. We have added these new results and analysis to appendix A.2.2
>
> Table 3: Downstream test AP Results with different K during training. Both CaW and DyG2Vec were run with optimal hyperparameters for each dataset.
>
> | Training $K$ |  Model  |        |  MOOC  |        |        |   UCI  |         |        |  Enron |        |
> |:------------:|:-------:|:------:|:------:|:------:|:------:|:------:|:-------:|:------:|:------:|:------:|
> |              |         |        |        |        |        |        |         |        |        |        |
> |              |         |   $1$  |  $200$ | $2000$ |   $1$  |  $200$ |  $2000$ |   $1$  |  $200$ | $2000$ |
> |     $K=1$    | DyG2Vec | $93.0$ | $84.3$ | $74.5$ | $97.2$ | $89.6$ | $84.03$ | $96.2$ | $91.9$ |  $90.3$  |
> |              |   CaW   | $94.0$ | $82.2$ | $73.2$ | $93.9$ | $85.6$ |  $76.6$ | $97.0$ | $92.9$ | $92.9$ |
> |    $K=200$   | DyG2Vec | $89.2$ | $87.8$ | $79.8$ | $95.3$ | $91.8$ |  $86.6$ | $91.1$ | $89.4$ | $87.9$ |
> |              |   CaW   | $81.9$ | $61.2$ | $60.8$ | $84.2$ | $63.6$ |  $58.0$ | $91.8$ | $78.6$ | $78.0$ |

---

> ### Author Response · Authors · 2022-11-15
> **Author Response Part 4**
>
> > 4.6 Missing important related works. There are a few related works about self-supervised dynamic graph learning methods, e.g [1-4].
>
> We thank the reviewer for pointing out these works. We were aware of these prior works; however, we chose not include them in the related work discussion as their goals do not align with our work. In general, none of these works study their methods under the common settings of the SSL framework as their goals are different. That is, they are concerned with improving the performance on known downstream tasks using auxiliary losses; there is no general pre-training mode. In light of the reviewer identifying these papers, we realize that it would be useful to have a discussion in the paper of the relationship with and difference from our work. We provide a thorough review of each paper below along with an empirical comparison. We have added these works and empirical results to Appendix A.8.
>
> DDGCL [1] adapts the TGAT encoder with a self-supervised contrastive framework across time. This extends the classic contrastive learning paradigm  by contrasting two nearby temporal views of the same node using a time-dependent similarity metric. Moreover, a debiased contrastive loss is used to correct the typical negative sampling bias in contrastive learning. Experiments on the fine-tune and multi-task learning settings show that the approach improves over the simple TGAT encoder both for future link prediction and dynamic node classification. Nonetheless, their approach comes with several shortcomings. First, it is built on the TGAT encoder which, as seen in Tables 1 and 5 in the paper, is a weak encoder, particularly for large datasets. Second, experiments are limited to the Reddit and Wikipedia datasets which are relatively easy. Last, the authors do not experiment under the standard settings in the graph SSL literature such as the freeze and semi-supervised settings. Table 4 below shows the results for downstream future link prediction under the freeze setting. The results show up to 10\% gap compared to DyG2Vec, particularly for datasets where the TGAT encoder under-performs (e.g. Enron, UCI).
>
> Table 4: Downstream Freeze test AP Results (after pre-training) for K=1. DDGCL pre-training and downstream training were run with default parameters described in the work. DyG2Vec was run with optimal parameters for each dataset.
>
> |  Model  |       MOOC      |      Enron      |       UCI       |      LastFM     |
> |:-------:|:---------------:|:---------------:|:---------------:|:---------------:|
> | DyG2Vec | $\mathbf{86.1}$ | $\mathbf{90.1}$ | $\mathbf{94.5}$ | $\mathbf{79.8}$ |
> |  DDGCL  |      $84.3$     |        83       |       85.3      |       78.8      |
>
> Cong. el al. [2] propose the dynamic graph transformer (DGT), which is a transformer-based graph encoder for **discrete-time dynamic graphs**. DGT is composed of two-tower networks that embed the temporal evolution and topological information of the input graph. Moreover, a temporal-union graph structure is proposed to efficiently summarize the temporal evolution into one graph. DGT is trained to encode the temporal-union graph using two complementary self-supervised pretext tasks, namely temporal reconstruction and multi-view contrasting. The first aims to reconstruct a snapshot given the past and present similar to how language models are trained. On the other hand, the latter is trained via non-contrastive learning on two views with randomly masked nodes. All together, DGT outperforms SOTA discrete-time  baselines on several datasets for link prediction tasks. While this work operates in a different domain (discrete time vs. continuous time), an interesting direction for future work would be to adapt the pre-training strategy for continuous-time dynamic graphs.
>
>
> [1] Tian, S., Wu, R., Shi, L., Zhu, L., and Xiong, T. 2021. Self-supervised Representation Learning on Dynamic Graphs. Proceedings of the 30th ACM International Conference on Information \& Knowledge Management.
>
> [2] Cong, W., Wu, Y., Tian, Y., Gu, M., Xia, Y., Mahdavi, M., and Chen, C.J. 2021. Dynamic Graph Representation Learning via Graph Transformer Networks. arXiv preprint arXiv:2111.10447.

---

> ### Author Response · Authors · 2022-11-15
> **Author Response Part 4 Cont.**
>
> > 4.6 "Missing important related works. There are a few related works about self-supervised dynamic graph learning methods, e.g [1-4].
>
> Wang et al. [3] adapt the vanilla transformer architecture to dynamic graphs by designing a two-stream encoder that extracts temporal and structural information from the temporal neighborhoods associated with any two interaction nodes. Rather than treating link prediction as a binary classification task, the authors leverage a contrastive learning strategy that maximizes the mutual information between the representations of future interaction nodes. Experiments show improved performance on future link prediction due to the more robust contrastive training strategy. However, the paper does not compare to the SoTA CaW method. Moreover, experiments are limited to the future link prediction task.
>
> Jiang et al. [4] adapt a sub-graph contrastive learning method [5] where a node representation is contrasted in both structure and time. Experiments on downstream link prediction task under the freeze setting show improvement over baselines. While interesting, the approach has several shortcomings. First, the initial node features are computed as one-hot encodings which makes the method unsuitable for the inductive scenario. Second, the contrastive learning method can result in high memory and computation due to the negative sampling [6]. This makes the method less desirable for large-scale graphs. Third, there are no results on other downstream tasks (e.g., dynamic node classification). Despite requests, we were unable to obtain code to conduct a performance comparison and implementation is non-trivial.
>
> [3] Wang, L., Chang, X., Li, S., Chu, Y., Li, H., Zhang, W., He, X., Song, L., Zhou, J., and Yang, H. 2021. TCL: Transformer-based Dynamic Graph Modelling via Contrastive Learning. arXiv preprint arXiv:2105.07944.
>
> [4] Jiang, L., Chen, K.J. and Chen, J., 2021. Self-Supervised Dynamic Graph Representation Learning via Temporal Subgraph Contrast. arXiv preprint arXiv:2112.0873.
>
> [5] Jiao, Y., Xiong, Y., Zhang, J., Zhang, Y., Zhang, T., and Zhu, Y. 2020. Sub-graph Contrast for Scalable Self-Supervised Graph Representation Learning. IEEE International Conference on Data Mining (ICDM).
>
> [6] Thakoor, S., Tallec, C. , Azar, M.G., Azabou, M. , Dyer, E.L., Munos, R., Velickovic, P., and Valko, M. 2022. Large-scale representation learning on graphs via bootstrapping. In Proceedings for the International Conference on Learning Representations.

---

> > ### Author Response · Authors · 2022-11-22
> > **Kind Request for Feedback**
> >
> > We thank the reviewer again for their insightful comments. To summarize, we tried to address the following concerns:
> >
> > 1. We clarify the motivation behind window-based SSL pre-training and provide experiments to show it's advantage.
> >
> > 2. We address the concern on novelty and provide further experiments to justify our choice of distortions.
> >
> > 3. We provide experiments to analyze the advantange of DyG2Vec over baselines in forecasting.
> >
> > 4. We discuss all suggested related works and add experiments to show the benefits of our method
> >
> > 5. We submitted a revised paper to improve notation and address all your concerns.
> >
> >
> > We would appreciate it if you have any further questions or concerns.

---

> > > ### Author Response · Authors · 2022-12-06
> > > **Do you have further concerns?**
> > >
> > > We thank the reviewer for their time and thoughtful review. Please let us know if anything requires further clarification.

---

### Official Review · Reviewer_fBK8 · 2022-10-25

**Confidence:** 4
**Correctness:** 3
**Technical Novelty And Significance:** 3
**Empirical Novelty And Significance:** 3
**Recommendation:** 6

**Clarity, Quality, Novelty And Reproducibility:**

The paper is well-written, and the problem is interesting. The paper is well-placed in the literature, and relevant works are cited and referenced. However, the experimental results are not too convincing. The idea is somewhat novel. The work would benefit the GNN community if the code is available.


**Strength And Weaknesses:**

1. The problem is challenging and impactful.
2. Being an SSL framework, it has more advantages than supervised models. It can be used for a wide variety of tasks.
3. Experimental results in Tables are not convincing although some of the results like Fig3 and Fig6 are interesting.
4. Paper is easy to follow, and well-written. However, I see a major weakness in experimental results.
5. Although there is a benefit of using DyG2Vec due to its inexpensiveness. However, it looks like CaW does a better job overall.
6. This method might be useful for large graphs if this can make training faster at the cost of some % of accuracy but cannot be considered for all the scenarios.
7. The authors mentioned in 5.1 that existing methods evaluate for K=1. However, this paper includes more results with K=200, 2000. What is the rationale for doing this? I see that results for K=1 are not good for DyG2Vec but better for K=200, 2000. Any intuition behind this is not explained. Why do other works have considered only K=1?
8. Fig6 shows that MOOC and UCI datasets are sensitive against the value of W.
9. I have seen a similar work [1] but for meshes in which the method takes a mesh and predicts a mesh for a future timestamp. Can you please comment on the similarity with this work?
[1] LEARNING MESH-BASED SIMULATION WITH GRAPH NETWORKS

**Summary Of The Paper:**

The paper proposes a novel SSL pretraining architecture based on encoder-decoder model called DyG2Vec for dynamic graphs. It is a fixed window-based method to learn node embeddings for future predictions. Paper uses two views of temporal subgraphs in a non-contrastive SSL framework. The SSL objective consists of three terms, invariance criterion, variance term and covariance term. Two evaluation mechanisms are adapted for experiments. Authors performed two downstream tasks such as future link prediction and dynamic node classification on 7 benchmark datasets.

**Summary Of The Review:**

The work is presented well and has advantages being an SSL framework, inexpensive, etc. However, I have a few concerns that it is not outperforming in the majority of the settings. Rather CaW looks more like a generalized method but it is quite expensive. What is the rationale behind using the K>1 setting? There is similar work in a different domain (meshes)

---

> ### Author Response · Authors · 2022-11-15
> **Author Response Part 1**
>
> Thank you for your valuable time and review! Please find our response below.
>
> > 3.3 "Experimental results in Tables are not convincing although some of the results like Fig3 and Fig6 are interesting.'', 3.4 "Although there is a benefit of using DyG2Vec due to its inexpensiveness. However, it looks like CaW does a better job overall.'', 3.5 "Paper is easy to follow, and well-written. However, I see a major weakness in experimental results.'' 3.6 "This method might be useful for large graphs if this can make training faster at the cost of some of accuracy but cannot be considered for all the scenarios".
>
> We address weaknesses 3.3-3.6 as a group because they all pertain to the experimental performance, and particularly the comparison with CaW. We would like to emphasize some important considerations with respect to performance. We have modified the introduction of the paper and the discussion of the results to make these aspects clearer.
>
> (1) CaW learns edge representations as opposed to node representations. This means that it is both limited to link-based tasks (primarily link prediction) and tailored to link prediction. In contrast, DyG2Vec can be used for node or link tasks. Arguably, SSL is more likely to be important in a node classification setting because, whereas one can always observe edges in a dynamic graph, acquiring node labels can be challenging. CaW is trained for the same task it is tested on. It is not clear that it learns a representation that is agnostic to the downstream task; it performs well on link prediction, but it is specifically designed for the link prediction task and trained and tested on that task. In this sense, it has an unfair advantage for the link prediction task. (3) CaW targets the one-step-ahead link prediction task (K=1). There are multiple practical settings where it is important to perform forecasts further into the future so that one has time to take an action in response to a prediction (e.g. online portfolio management in dynamic financial networks). While CaW outperforms DyG2Vec for 5/7 datasets for K=1 for transductive link prediction, DyG2Vec outperforms CaW for 5/7 datasets for K=200 and 4/7 for K=2000 (with equal performance for Reddit). The training for DyG2Vec uses a loss function associated with the K=200 case for all 3 settings. We can improve performance on the K=1 case considerably by using a K=1 loss function tailoring for this task (See Appendix A.2.2). On the other hand, CaW cannot easily be improved for the longer forecast horizons (our efforts lead to worse performance). If the huge discrepancy in computational requirements is not important for some setting, then we acknowledge that CaW is a superior choice for the inductive link prediction task.
>
> > 3.7: The authors mentioned in 5.1 that existing methods evaluate for K=1. However, this paper includes more results with K=200, 2000. What is the rationale for doing this? I see that results for K=1 are not good for DyG2Vec but better for K=200, 2000. Any intuition behind this is not explained. Why do other works have considered only K=1?
>
> In this paper, we introduce the Next-K evaluation setting to provide better insight into the forecasting abilities of the models for dynamic graph models. Hopefully, this will encourage the community to start looking into longer-range forecasting for dynamic graphs. In most other dynamic settings (e.g. multivariate time series), longer horizon forecasting is a key task that serves as a performance benchmark. An ideal model with good forecasting abilities can help make more informed decisions about the future, beyond the next immediate event. An example is online portfolio management in a dynamic transaction network [1]. This is analogous to time-series forecasting [2] where models are trained to predict trends over a certain horizon, and trajectory prediction in autonomous driving [3] where models predict the trajectory of a car over the next couple of seconds before making a planning decision.
>
> [1] Wei, W., Zhang, Q., and Liu, L. 2021. Bitcoin Transaction Forecasting With Deep Network Representation Learning. IEEE Transactions on Emerging Topics in Computing.
>
> [2] Lim, B., and Zohren, S. 2021. Time-series forecasting with deep learning: a survey. Philosophical Transactions of the Royal Society.
>
> [3] Liang, M., Yang, B., Hu, R., Chen, Y., Liao, R., Feng, S., and Urtasun, R. 2020. Learning Lane Graph Representations for Motion Forecasting. ECCV.

---

> ### Author Response · Authors · 2022-11-15
> **Author Response Part 2**
>
> > 3.8 Fig6 shows that MOOC and UCI datasets are sensitive against the value of W
>
> Response 3.8: Thank you for raising this issue. The y-axis of Figure 6 perhaps gives a misleading indication of significant sensitivity. Performance of UCI ranges from 0.92-0.95 for the entire range of window sizes (512-64K). Similarly, MOOC performance ranges from 0.86-0.90 over the same range. We would argue that a 4 percent range is not indicative of extreme sensitivity. More importantly, in the more practical range of window sizes (2K-16K), which are long enough to include sufficient historical information but not too long so that they start to drown the model with irrelevant data, the performance varies over the ranges 0.94-0.95 and 0.895-0.905, respectively. This is only a one percent range in each case. We can thus choose any window size in this practical set and achieve close to optimal performance. Larger values incur a greater training time and memory overhead, so 8K is a reasonable trade-off. We have not encountered datasets that require extremely large windows, but such dynamic graphs (e.g. exhibiting periodic/seasonal behavior
> with a very long period) may need an alternative multi-scale approach, and this is an interesting research question, but not within the scope of
> our paper.
>
>
> > 3.9 I have seen a similar work [1] but for meshes in which the method takes a mesh and predicts a mesh for a future timestamp. Can you please comment on the similarity with this work? [1] LEARNING MESH-BASED SIMULATION WITH GRAPH NETWORKS
>
> We thank the reviewer for bringing this work to our attention. We provide a brief discussion of it below. We acknowledge that there are connections with our work, but there are also major differences.
>
> While this work does involve predicting the future given a window of the past, it does not operate on continuous-time dynamic graphs but rather meshes. In addition, the task is not to predict the future interactions in a graph but rather future dynamical quantities $q_i$ for each node $i$ in the next time step. Therefore, it can be thought of as a multi-variate time-series forecasting problem or a discrete-time dynamic graph with evolving node features. Time is not encoded explicitly in the history but instead  message passing is done over a static multi-graph. Unlike our setting, the windows are very short (window size $<5$ where 1 works best).
>
> > 3.Summary: Rather CaW looks more like a generalized method but it is quite expensive.
>
> We would argue that CaW is in fact the least general of the baselines. CaW is specifically optimized for the link prediction task. It does not generate node representations and cannot be used for node-related downstream tasks. Experiments with CaW, both in the original paper and our paper, focus only on the link prediction task, so there is no examination of whether CaW can generate representations that are useful for tasks that are different from the training task (even other link-based tasks such as edge classification).

---

> > ### Author Response · Authors · 2022-11-22
> > **Kind Request for Feedback**
> >
> > We thank the reviewer again for their insightful comments. To summarize, we tried to address the following concerns:
> >
> > 1. We clarified the advantages of DyG2Vec over SoTA baselines.
> >
> > 2. We justify the forecasting evaluation settings provided in the paper.
> >
> > 3. We address the concern on sensitivity to window size.
> >
> > 4. We discuss the suggested related work and it's relation to continuous-time dynamic graphs.
> >
> > 5. We clarify the limits of the CaW method.
> >
> > We would appreciate it if you have any further questions or concerns.

---

> > > ### Author Response · Authors · 2022-12-06
> > > **Do you have further concerns?**
> > >
> > > We thank the reviewer for their time and thoughtful review. Please let us know if anything requires further clarification

---

> > > > ### Comment · Reviewer_fBK8 · 2022-12-07
> > > > **Thank you**
> > > >
> > > > Thank you for the detailed clarification. I would like to keep my rating the same.

---

### Official Review · Reviewer_gYTF · 2022-11-03

**Confidence:** 3
**Correctness:** 3
**Technical Novelty And Significance:** 2
**Empirical Novelty And Significance:** 3
**Recommendation:** 6

**Clarity, Quality, Novelty And Reproducibility:**

The paper's clarity is good - most of the text would be understood conveniently to the readers.  In terms of quality, the paper presents a method that is intuitive and considers multiple factors when learning on dynamic graphs, such as temporal encodings, window based training, etc. Although there are a few issues (see weaknesses above), overall the clarity and quality is sufficient. In terms of originality, the paper presents a method that is motivated by existing works but used for SSL representation learning on dynamic graphs. In my evaluation, the SSL pre-training, window based training and temporal edge encoding are the new additions to the architecture-- some of which are motivated from prior works. Finally, there is no theoretical investigation of DyG2Vec -- refer [2] for a related work.

**Strength And Weaknesses:**

Strengths:
1. The paper contributes an approach that can be used to perform SSL pre-training on dynamic graphs using non-contrastive approach. This evidently improves the trained architecture for temporal graph tasks in a majority of the empirical evaluations shown.
2. The background of the research topic is clearly written which follows the need to proposed DyG2Vec.
3. The window-based training is interesting and from the experiments its observed that it is able to predict longer future events.
4. Figure 3 study shows DyG2Vec is more efficient yet high performing compared to baselines.
5. Ablation studies and experimental details are sufficiently mentioned that provides robustness of the studies as well as details for reproducibility later (after code release).

Weaknesses and questions:
1. There are couple of related works that could be discussed - SSL for dynamic graphs using contrastive loss [1], and provably expressive temporal networks [2], and where possible, their performance comparison with DyG2Vec can be mentioned.
2. Page 6: How is temporal degree centrality is computed for an edge? It mentions "the current degrees of nodes u_p and v_p at time t_p." the degrees of u_p and v_p could be different; how are the two values combined to make the edge degree centrality?
3. After reading the experimental discussion, the curiosity on what is the major contribution of DyG2Vec for the improve in performance -- SSL pre-training? window-based framework? temporal edge encoding? For instance, if its SSL pre-training can it be easily used in prior models as an augmentation or extension and whether it would provide an equivalent boost in performance? And for temporal edge encoding -- is it providing features to the architecture which otherwise cannot be learned by the temporal graph attention?
4. Page 8: "Table 2 shows that DyG2Vec outperform all baselines significantly" - Although the performance of DyG2Vec is impressive, the quoted statement in Page 8 (as well as similar claims in other experimental sections) is misleading and incorrect as per the table of results. For example, in Table 2, DyG2Vec does not beat baselines in 2 out of 9 instances. Such precise numbers may be mentioned to correct the claims as quoted.
5. For the performance versus Inference time graphs in Figure 3, how do the models compare in terms of parameters or sizes? Are all the models plotted in Figure 3 are of comparable model size (eg. number of layers or number of parameters)?

[1] Jiang, L., Chen, K.J. and Chen, J., 2021. Self-Supervised Dynamic Graph Representation Learning via Temporal Subgraph Contrast.
[2] Souza, A.H., Mesquita, D., Kaski, S. and Garg, V., 2022. Provably expressive temporal graph networks.

**Summary Of The Paper:**

This paper proposes a method to improve representation learning for dynamic graphs using self-supervised learning (SSL) approach. Named DyG2Vec, the method is equipped with a SSL pre-training component that uses a non-contrastive loss objective, a window-based architecture as well as fine-tuning component that can tune the task-agnostic representations for the concerned dynamic graph learning tasks. As for the architecture, an encoder decoder pipeline is adopted that uses attention based graph neural network, as well as additional temporal edge encoding which captures degree centrality as well as common neighbors between two nodes. Experimental evaluation on multiple datasets show promise of using DyG2Vec.

**Summary Of The Review:**

The paper overall does great in presenting a self-supervised pre-training approach that can be used to generate task-agnostic node representations for dynamic graph learning, and is intuitively and empirically powerful than existing works. There are a few missing details (as asked in the weaknesses and questions section above) which can further be provided or answered.

---

> ### Author Response · Authors · 2022-11-15
> **Author Response Part 1**
>
> Thank you for your valuable time and review! Please find our response below.
>
> > 2.1 There are couple of related works that could be discussed - SSL for dynamic graphs using contrastive loss [1], and provably expressive temporal networks [2], and where possible, their performance comparison with DyG2Vec can be mentioned.
>
> Response 2.1: We thank the reviewer for pointing out these works. We have added a brief summary of each work to the revised version of the paper (See Appendix A.8). Despite requests, we were unable to obtain code for [1] to conduct a performance comparison and implementation is non-trivial. The paper [2] is a very recent work (published Sept. 29 2022), so performance comparison was not feasible for the submission.
>
> Jiang et. al [1] adapt a sub-graph contrastive learning method [6] where a node representation is contrasted in both structure and time. Experiments on downstream link prediction task under the freeze setting show improvement over baselines. While interesting, the approach has several shortcomings. First, the initial node features are computed as one-hot encodings which makes the method unsuitable for the inductive scenario. Second, the contrastive learning method can result in high memory and computation due to the negative sampling [3]. This makes the method less desirable for large-scale graphs. Third, there are no results on other downstream tasks (e.g., dynamic node classification).
>
> Souza et al. [2] is a recent work (released Sept. 29 2022 on Arxiv) that establishes a series of useful theoretical results concerning temporal graph encoders. The analysis exposes several weakness of both memory-based methods (e.g., TGN) and walk-based methods (e.g., CaW). Given these insights, the authors propose PINT, a memory-based method that leverages injective message-passing and novel relative positional encodings.
> Experiments show significant improvement over SoTA baselines on the link prediction task. An interesting direction for future research would be to evaluate the expressive power of DyG2Vec compared to baselines using their theoretical framework (e.g., temporal WL test).
>
>
> [1] Jiang, L., Chen, K.J. and Chen, J., 2021. Self-Supervised Dynamic Graph Representation Learning via Temporal Subgraph Contrast. arXiv preprint arXiv:2112.0873.
>
> [2] Souza, A.H., Mesquita, D., Kaski, S. and Garg, V., 2022. Provably expressive temporal graph networks. Advances in Neural Information Processing Systems.
>
> [3] Thakoor, S., Tallec, C. , Azar, M.G., Azabou, M. , Dyer, E.L., Munos, R., Velickovic, P., and Valko, M. 2022. Large-scale representation learning on graphs via bootstrapping. In Proceedings for International Conference on Learning Representations.
>
> > 2.2 Page 6: How is temporal degree centrality is computed for an edge? It mentions 'the current degrees of nodes $u_p$ and $v_p$ at time $t_p$.' the degrees of $u_p$ and $v_p$ could be different; how are the two values combined to make the edge degree centrality?
>
> Response 2.2: We apologize for the confusion. The current degrees of nodes $u_p$ and $v_p$ at time $t_p$ are simply concatenated together, along with the the number of common 1-hop neighbors to get the vector $\Theta_p(t_p) \in \mathbb{R}^3$. This has been clarified in the main text of the revised paper (See page 6 under equation 9).
>
> > 2.3(a) After reading the experimental discussion, the curiosity on what is the major contribution of DyG2Vec for the improve in performance -- SSL pre-training? window-based framework? temporal edge encoding?
>
> We thank the reviewer for raising this question. All the aforementioned points contribute to the performance of DyG2Vec. The window-based framework provides a natural bias towards recent edges. When combined with uniform-based sampling, this helps capture both recent and recurrent interactions which are highly indicative of future interactions. Moreover, having a limited window helps alleviate the common over-squashing and over-smoothing issues for large graphs. For example, Fig. 6 in our main paper shows that increasing the window size to 64K for UCI (which is the same as having no window since UCI has 60K edges) results in a 4\% drop in performance. When removing temporal edge encodings, we have an additional 1\%-5\% drop in performance. We have made these clarifications in Section 5.3.
>
> The SSL framework is primarily helpful for settings where task labels are sparse or highly imbalanced (See next comment).

---

> ### Author Response · Authors · 2022-11-15
> **Author Response Part 2**
>
> > 2.3(b) For instance, if its SSL pre-training can it be easily used in prior models as an augmentation or extension and whether it would provide an equivalent boost in performance?
>
> SSL pre-training is particularly helpful in transfer learning scenarios and cases where task labels are sparse and/or highly imbalanced.
>
> This arises in anomaly detection tasks which are common in dynamic graphs. For example, for the dynamic node classification datasets Reddit (0.05\%), Wikipedia (0.1\%), and MOOC (1\%), fewer than 1\% of interactions result in a ban (positive label). Our results in Fig. 5 in the paper show up to 5\% improvement on downstream performance with SSL pre-training versus link prediction pre-training. Unfortunately, as mentioned in the paper, it is challenging to adapt prior art to the standard graph SSL paradigm. For example, memory-based models require sequentially processing a dynamic graph to build the memory and generate temporal embeddings. Therefore, any random view of the dynamic graph will need to be sequentially encoded which is expensive. In general, the prior methods do not generate task-agnostic node embeddings that can be SSL pre-trained.
>
> > 2.3(c)And for temporal edge encoding -- is it providing features to the architecture which otherwise cannot be learned by the temporal graph attention?
>
> Temporal edge encodings provide inductive bias which is otherwise hard to learn with pure attention. This is, in part, why pure attention models (e.g., TGAT) fail on large  datasets like LastFM. See the 30\% performance gap in Table 1. Moreover, our ablation studies in Fig. 6 show up to 6\% gap when removing the temporal edge encodings.
>
> > 2.4 Page 8: "Table 2 shows that DyG2Vec outperform all baselines significantly" - Although the performance of DyG2Vec is impressive, the quoted statement in Page 8 (as well as similar claims in other experimental sections) is misleading and incorrect as per the table of results. For example, in Table 2, DyG2Vec does not beat baselines in 2 out of 9 instances. Such precise numbers may be mentioned to correct the claims as quoted.
>
> We thank the reviewer for raising this valid concern. All such claims have been modified in the revised version. We have eliminated all general claims about outperforming and now characterize the concrete performance (See the first and second paragraph in Page 8 of our paper).
>
> > 2.5 For the performance versus Inference time graphs in Figure 3, how do the models compare in terms of parameters or sizes? Are all the models plotted in Figure 3 are of comparable model size (eg. number of layers or number of parameters)?
>
> Overall, all baselines are of similar model size and number of layers (See Appendix A.6). For the baselines, we adopt the optimal hyperparameters determined by previous work. The inference time difference is mainly attributed to how the baselines process the input graph to predict a batch of target edges. For example, CaW encodes $M$ walks for each target edge separately, while DyG2Vec encodes a single subgraph for all target edges and operates on a fixed window size. We have added further details in Appendix A.2.3.

---

> > ### Comment · Reviewer_gYTF · 2022-11-18
> > **Thank you**
> >
> > Thank you for your clarifications on the questions in the initial review and the corresponding revisions in the paper.

---

> > > ### Author Response · Authors · 2022-11-22
> > > **Kind Request for Feedback**
> > >
> > > We thank the reviewer again for their insightful comments. To summarize, we tried to address the following concerns:
> > >
> > > 1. We provide a discussion on the suggested related works and add them to the latest revision.
> > >
> > > 2. We clarify the major contributions of DyG2Vec in terms of performance and efficiency.
> > >
> > > 3. We clarified that all models are of comparable size and added time complexity analysis to latest revision.
> > >
> > > We would appreciate it if you have any further questions or concerns.

---

> > > > ### Author Response · Authors · 2022-12-06
> > > > **Do you have further concerns?**
> > > >
> > > > We thank the reviewer for their time and thoughtful review. Please let us know if anything requires further clarification

---

### Official Review · Reviewer_46gJ · 2022-11-04

**Confidence:** 4
**Clarity, Quality, Novelty And Reproducibility:** Please see the above comments.
**Correctness:** 3
**Technical Novelty And Significance:** 2
**Empirical Novelty And Significance:** 2
**Recommendation:** 5

**Strength And Weaknesses:**

The idea proposed in this paper is interesting to me, and experiments show some empirical improvements and efficiency. However, I still have the following concerns and questions for the current version of this paper:

1. Section 4 is not easy to follow. In particular, the notation definitions are not clear enough. For example, at the beginning of Section 4, Eq (2) has M+1 intervals rather than M; What's the difference between stride S and window length W? Can the intervals defined in Eq (2) overlap?

2. Window size is essential to capture the temporal pattern and varies on different datasets (according to Figure 6 left). It is difficult to predefine, which hinders the practical use of the proposed approach.

3. The experimental results are mixed. The baseline CAW does better in many settings, especially on the inductive link prediction task (Table 5).

4. I'm curious about the efficiency comparison against CAW because CAW proposes an efficient link sampling algorithm and claims high efficiency. In addition to Figure 3, it would be better to give some formal time complexity analysis or profiling. This will make readers understand where the gap is.

**Summary Of The Paper:**

This paper proposes a self-supervised learning framework for dynamic graphs. Temporal edges are split into windows which are then used as sub-graph samples for self-supervised learning. The proposed model is efficient and obtains empirical improvements on several datasets.

**Summary Of The Review:**

An interesting approach for dynamic graphs but further investigations are needed.

---

> ### Author Response · Authors · 2022-11-15
> **Response Part 1**
>
> Thank you for your valuable time and review! Please find our response below.
>
> > 1.1 Section 4 is not easy to follow. In particular, the notation definitions are not clear enough. For example, at the beginning of Section 4, Eq (2) has M+1 intervals rather than M; What's the difference between stride S and window length W? Can the intervals defined in Eq (2) overlap?
>
> Response 1.1:    We apologize for the lack of clarity of Section 4. In the revised version of the paper, we have revisited the section. We have ensured that all notation is clearly defined. (a) The number of intervals should be $M$; the $M+1$ was a typo and has been corrected. (b) We have added a sentence to explain the difference between the stride $S$ and the window length $W$. The stride is used only for training purposes; successive mini-batches during training are separated by $S$. The window length determines how many historical edges are used to perform forecasting. In our experiments, we set $S=K$ so that each edge is only predicted once in each epoch. We explore performance for different choices of $W$, but observe that the fixed choice $W=8K$ performs well for all datasets and forecasting tasks.
>
> > 1.2 Window size is essential to capture the temporal pattern and varies on different datasets (according to Figure 6 left). It is difficult to pre-define, which hinders the practical use of the proposed approach.
>
> Response 1.2: We thank the reviewer for raising this concern. As shown in Table 8 in the paper, the optimal window size for each dataset is indeed different due to distinct temporal patterns. This observation aligns with previous work [1] where it was found that temporal motifs develop over different timescales for different datasets.
>
> Despite this, we note that our results in Figure 6 of the paper show that a constant window size of 8K results in performance that is within 1-3\% of optimal performance for every dataset. The performance range for window sizes in the practical set of 2K-16K is small (on the order of 1\% for most datasets). The attention-based architecture, combined with the Time2Vec module (see Eq. 8), helps the model be aware of the relative time-span between interactions. Therefore, the temporal attention can learn whether to attend more to recent or far interactions within the window.
>
> As a result, in a practical setting, we can choose a window size of 8K and be confident of achieving close-to-optimal performance. We have not encountered datasets that require extremely large windows, but such dynamic graphs (e.g. exhibiting periodic/seasonal behaviour with a very long period) may need an alternative multi-scale approach, and this is an interesting research question, but not within the scope of our paper.
>
> > 1.3 "The experimental results are mixed. The baseline CAW does better in many settings, especially on the inductive link prediction task (Table 5)."
>
> Response 1.3: Thank you for raising this concern. We admit that CaW is a probably a superior method for inductive link prediction due to its random temporal walk sampling procedure. Since several reviewers raised similar concerns, we have discussed the experimental performance of the proposed method (especially with respect to CaW) in a general comment (please see above). Here, we emphasize three points: (1) CaW is limited to link-based tasks and cannot be used for the arguably more SSL-relevant task of node classification; (2) it is not clear that CaW forms task-agnostic representations - it is trained and tested on the link prediction task, both in the original paper and in our work. By contrast, all other baselines are also tested in the node classification setting while being trained on the link prediction task; (3) DyG2Vec is 40-100x faster than CaW (as shown in Figures 3 and 4 of our main paper), making it more suitable for streaming settings.
>
> [1] Paranjape, A., Benson, A.R., and Leskovec, J. 2017. Motifs in Temporal Networks. Proceedings of the Tenth ACM International Conference on Web Search and Data Mining.

---

> ### Author Response · Authors · 2022-11-15
> **Response Part 2**
>
> > 1.4 I'm curious about the efficiency comparison against CAW because CAW proposes an efficient link sampling algorithm and claims high efficiency. In addition to Figure 3, it would be better to give some formal time complexity analysis or profiling. This will make readers understand where the gap is.
>
> Response 1.4:  We note that all CaW experiments use the proposed efficient sampling. However, the main overhead lies in how each of the baselines processes the input graph to predict a batch of $K$ target edges. CaW samples $M$ $L$-hop random walks for each target edge. This is followed by an expensive set-based anonymization scheme. To achieve good performance, CaW can require relatively long walks (e.g., for Enron, $L=5$). On the other hand, memory-based methods and TGAT sample a different $L$-hop subgraph for each target edge. DyG2Vec samples a single $L$-hop subgraph within a constant window size $W$ for all target edges. In other words, DyG2Vec benefits from a unified and shared message-passing procedure to predict all target edges. On contrary, the baselines perform disjoint computations for each target edge. The complexities for encoding are thus: DyG2Vec = $O(LW)$; CaW = $O(LKN_sM)$ and TGN and variants = $O(LKN_s)$. Here, $N_s$ represents the maximum number of sampled nodes in a L-hop subgraph and $K$ the number of target edges to predict. We can see that the main difference is the factor $M$ and the fact that a computationally expensive procedure is performed for each of the $M$ samples. The factor $N_s$ comes from the complexity of message passing at each hop for each node (assuming sparse operations). Note that DyG2Vec is limited to $O(W)$ nodes so it does not have this factor. We have added this discussion in the revised version of the paper (See Appendix A.2.3). Additionally, Table 1 below shows that doubling the window size results only in a near linear increase in runtime.
>
> Table 1: Effect of window size W on inference time (s) on the Wikipedia dataset. Inference time represents the time it takes to predict the whole test set.
>
> |   Window Size W  |   512  |   2K  |  4K  |  8K  |  16K |  32K |  64K | 128K |
> |:----------------:|:------:|:-----:|:----:|:----:|:----:|:----:|:----:|:----:|
> | Test Runtime (s) | 11.14, | 11.12 | 12.7 | 19.3 | 16.3 | 19.8 | 25.2 | 29.6 |

---

> > ### Author Response · Authors · 2022-11-22
> > **Kind Request for Feedback**
> >
> >
> > We thank the reviewer again for their insightful comments. To summarize, we tried to address the following concerns:
> >
> > 1. We clarify the advantages of DyG2Vec over the SoTA baselines.
> >
> > 2. We address the concern on sensitivity to window size.
> >
> > 3. We provide time complexity analysis and conduct experiments to show efficiency with respect to window size.
> >
> > 4. We submitted a revised paper to improve notation and address your suggestions.
> >
> > We would appreciate it if you have any further questions or concerns.

---

> > > ### Author Response · Authors · 2022-12-06
> > > **Do you have further concerns?**
> > >
> > > We thank the reviewer for their time and thoughtful review. Please let us know if anything requires further clarification

---

> ### Comment · Reviewer_46gJ · 2022-12-12
> **Thanks for response**
>
> Thanks for the authors' response and revision, but my concerns (especially points 2&3) are not fully resolved. Other reviewers also raised similar issues. Overall I will keep my rating.

---

### Author Response · Authors · 2022-11-15
**General Response: Performance compared to CaW**

We thank the reviewers for their insightful comments. We have taken all of them into account and they have help us significantly improve the revised version of the paper. Below, we make a general reply to all reviewers in order to address common points. Individual replies are provided to each reviewer to address specific issues raised in their review.

While we admit that we do not outperform CaW in all settings, we emphasize that there are some critical aspects of outperformance:
(1) Unlike other baselines, CaW is specifically tailored to the link prediction task and does not compute task-agnostic node embeddings. As a result, it cannot even be used for node classification (Table 2 in our main paper). This is arguably the more interesting task for self-supervised learning. As noted by Reviewer 4, for the link prediction task, the labels (links) are available, so we are essentially addressing a supervised task. On the other hand, node labels can often be scarce, because they require manual annotation in multiple settings (e.g., fraud detection in blockchain transaction networks [1]). In Table 2, DyG2Vec is not always the best method, but it achieves the best performance in 7 out of 9 tasks, and in several of these it is 10-15\% better than any other method.
(2) DyG2Vec is about 40-100x faster than CaW and slightly faster than other memory-based methods such as TGN (as shown in Figures 3 and 4 of our main paper). This is because the CaW sampling procedure must be performed for each target edge. The speed disadvantage makes CaW less desirable for streaming settings where real-time decision making is key, e.g., detecting anomalies/fraud on dynamic social/transaction networks. In addition, DyG2Vec comes with memory advantages as only a fixed history of edges are needed for future predictions. In contrast, baseline methods either assume access to the entire history or store a memory vector for each encountered node (e.g., TGN).
(3) CaW requires relatively careful tuning of the temporal bias $\alpha$ and the length of the walks $m$. With slightly different choices, performance can drop by 10\%. By contrast, DyG2Vec is more robust to hyper-parameter choice and can use a fixed window size of 8K for all datasets and tasks and achieve within 1-3\% of its optimal performance.


[1] Kim, J., Nakashima, M., Fan, W., Wuthier, S., Zhou, X., Kim, I., and Chang, S. 2022. A Machine Learning Approach to Anomaly Detection Based on Traffic Monitoring for Secure Blockchain Networking. IEEE Transactions on Network and Service Management.

---

### Decision · Program_Chairs · 2023-01-20

**Decision:**

Reject

**Justification For Why Not Higher Score:**

N/A

**Justification For Why Not Lower Score:**

N/A

**Metareview: Summary, Strengths And Weaknesses:**

This paper introduces DyG2Vec, a new window-based encoder-decoder model for dynamic graphs by using an attention-based message-passing mechanism that utilizes hierarchical multi-head attention modules to encode node embeddings across time. Furthermore, the authors present a joint-embedding architecture for dynamic graphs in which two views of temporal sub-graphs are encoded to minimize a non-contrastive loss function.

In general, the paper is well-written and easy to follow. The idea of the paper is interesting. However, the reviewers had some concerns about the contributions, related work, and experimental results and analysis of the paper. The authors addressed some of them but the reviewers are not fully convinced.